# The Histomorphology to Molecular Transition: Exploring the Genomic Landscape of Poorly Differentiated Epithelial Endometrial Cancers

**DOI:** 10.3390/cells14050382

**Published:** 2025-03-05

**Authors:** Thulo Molefi, Lloyd Mabonga, Rodney Hull, Absalom Mwazha, Motshedisi Sebitloane, Zodwa Dlamini

**Affiliations:** 1Discipline of Obstetrics and Gynaecology, School of Clinical Medicine, University of KwaZulu-Natal, Durban 4002, South Africa; thulo.molefi@up.ac.za; 2SAMRC Precision Oncology Research Unit (PORU), DSI/NRF SARChI Chair in Precision Oncology and Cancer Prevention (POCP) Pan African Research Institute (PACRI), University of Pretoria, Hartfield, Pretoria 0028, South Africa; 3Department of Medical Oncology, University of Pretoria, Hatfield, Pretoria 0028, South Africa; 4Department of Anatomical Pathology, National Health Laboratory Services, Durban 4058, South Africa

**Keywords:** endometrial cancer, poorly differentiated epithelial tumors, histomorphology, molecular classification, genomic profiling, targeted therapy, precision oncology

## Abstract

The peremptory need to circumvent challenges associated with poorly differentiated epithelial endometrial cancers (PDEECs), also known as Type II endometrial cancers (ECs), has prompted therapeutic interrogation of the prototypically intractable and most prevalent gynecological malignancy. PDEECs account for most endometrial cancer-related mortalities due to their aggressive nature, late-stage detection, and poor response to standard therapies. PDEECs are characterized by heterogeneous histopathological features and distinct molecular profiles, and they pose significant clinical challenges due to their propensity for rapid progression. Regardless of the complexities around PDEECs, they are still being administered inefficiently in the same manner as clinically indolent and readily curable type-I ECs. Currently, there are no targeted therapies for the treatment of PDEECs. The realization of the need for new treatment options has transformed our understanding of PDEECs by enabling more precise classification based on genomic profiling. The transition from a histopathological to a molecular classification has provided critical insights into the underlying genetic and epigenetic alterations in these malignancies. This review explores the genomic landscape of PDEECs, with a focus on identifying key molecular subtypes and associated genetic mutations that are prevalent in aggressive variants. Here, we discuss how molecular classification correlates with clinical outcomes and can refine diagnostic accuracy, predict patient prognosis, and inform therapeutic strategies. Deciphering the molecular underpinnings of PDEECs has led to advances in precision oncology and protracted therapeutic remissions for patients with these untamable malignancies.

## 1. Introduction

In 2023, it was estimated that there were 480,000 new endometrial cancer cases and 97,000 deaths among women, with endometrial cancer contributing to 3.2% of these deaths and 2.1% of new cases, making it the fourth most prevalent cancer in women [1]. Alarmingly, the incidence of endometrial cancer (EC) continues to grow worldwide, as highlighted in Figure 1. In the last 30 years, since the 1990s, the overall incidence of endometrial cancer has increased by 132%. The current belief is that this increase in incidence reflects an increase in the prevalence of risk factors for the disease, particularly higher rates of obesity and an aging population. In the United States, the incidence rate has increased by approximately 24% over 23 years and is projected to increase by an additional 35% by 2030 [2]. EC rates are highest in North America, followed by Eastern and Central Europe [3]. While the increase in diagnoses spans all age groups, the number of cases in women under 40 years of age has notably doubled, now comprising 4.2% of all low-grade endometrial cancer cases in the U.S [2]. Although high-income countries (HICs) have experienced the most significant rise in caseloads, age-standardized incidence rates are also increasing globally, including in sub-Saharan Africa [3]. Despite advances in treatment technologies, the mortality rates of endometrial cancer remain on the rise. Over the past 15 years, deaths related to endometrial cancer have increased by approximately 15%, underscoring the critical need for revising and enhancing treatment strategies [4].

From 1983, two main types of endometrial cancers were identified: Type I and Type II [5]. Type I endometrial cancers, comprising 70–80% of cases, are primarily endometrioid carcinomas (EECs). These tumors usually present at an early stage, are low-grade, and are associated with a favorable prognosis, achieving a five-year survival rate of over 85%. They are hormone-driven and closely linked to obesity and other components of metabolic syndrome [2]. Conversely, Type II endometrial cancers, herein referred to as poorly differentiated epithelial endometrial cancers (PDEECs), represent approximately 20% of cases. They include high-grade non-endometrioid subtypes, such as uterine papillary serous carcinomas (10%), clear cell carcinomas (1–5%), and carcinosarcomas (2–5%) [6]. Traditionally, ECs are classified histologically into two main types: endometrioid and non-endometrioid. The non-endometrioid category includes serous, clear cell, and undifferentiated carcinomas, among others [7]. PDEECs are predominantly found within this category, where they are associated with higher grades, deeper myometrial invasion, and an increased risk of metastasis compared with their well-differentiated counterparts [2]. They represent a subset of endometrial malignancies characterized by aggressive behavior, heterogeneous histopathological features, and distinct molecular profiles [8]. These cancers pose significant clinical challenges due to their propensity for rapid progression and poor response to conventional treatments and account for most endometrial cancer-related deaths [9].

PDEECs are often detected at more advanced stages and exhibit a higher risk of recurrence and mortality, even when diagnosed early [8]. They have limited treatment options and a five-year survival rate ranging between 36% and 80%. Currently, there are no targeted therapies for PDEECs, and they are still being managed in the same way as clinically indolent and readily curable type I ECs [4]. Therefore, new therapeutic approaches for the treatment of PDEECs are required. In this context, deciphering their unique characteristics remains crucial for improving diagnostic accuracy, treatment strategies, and patient outcomes [10]. Histologically, PDEECs are characterized by architectural disarray, marked cellular atypia, high mitotic index, and frequent tumor necrosis [2]. These features reflect the aggressive nature of the disease and contribute to the challenges in accurate diagnosis and prognosis determination based solely on histopathology. Nevertheless, from a molecular perspective, advancements in genomic profiling have illuminated the underlying genetic alterations driving PDEECs. Genetic alterations disrupt key cellular pathways involved in cell cycle regulation, apoptosis, DNA repair, and cell signaling, contributing to tumor initiation, progression, and therapeutic resistance [10]. The clinical significance of understanding PDEECs lies in their implications for personalized treatment approaches. Molecular profiling not only aids in the subclassification of tumors but also guides targeted therapies tailored to specific genetic alterations. Moreover, this underscores the importance of integrating genomic data with traditional histomorphological assessments to refine diagnostic accuracy and prognostic stratification [11].

Thus, elucidating the histomorphological landscape of PDEECs is essential for advancing our knowledge of their pathogenesis, refining clinical management strategies, and improving patient outcomes [8]. In this review, we explore the genomic landscape of PDEECs and how the transition from histopathology to molecular classification has provided critical insights into the underlying genetic and epigenetic alterations in these malignancies. This review also highlights the molecular alterations that drive the aggressiveness of these cancers and seeks to illuminate their implications in diagnosis, prognosis, and emerging treatment strategies. We discuss how molecular classification correlates with clinical outcomes and can refine diagnostic accuracy, predict patient prognosis, and inform therapeutic strategies.

## 2. Histological Classification of PDEECs

PDEECs are typically classified histologically based on their cell morphology and degree of cellular differentiation, with a focus on architectural and cytological features (Figure 2). These cancers belong to the high-grade or aggressive category of endometrial carcinomas [12]. The histologic subtypes of PDEECs are shown in Table 1, comprising high-grade endometrioid carcinoma, serous carcinoma, clear cell carcinoma, undifferentiated carcinoma, dedifferentiated carcinoma, and carcinosarcoma (malignant mixed Müllerian tumor) [13]. High-grade endometrioid carcinomas are tumors that are poorly differentiated, exhibiting >50% solid non-glandular, non-squamous growth or architecturally FIGO grade 2 endometrioid carcinoma that exhibits marked cytologic atypia [14]. Despite retaining some features of endometrioid carcinoma, these cancers exhibit aggressive clinical behavior and are more likely to exhibit lymphovascular invasion and metastasis [9]. Serous carcinoma is an aggressive form of endometrial cancer that typically presents with high-grade features. It accounts for a smaller percentage of endometrial cancers but is responsible for a disproportionate number of relapses and deaths [15]. Histologically, serous carcinomas are characterized by complex papillary structures, slit-like glands, high-grade nuclear atypia, and frequent mitosis. TP53 mutations are a hallmark of serous carcinoma, and these tumors often lack hormone receptor expression [16].

Clear cell carcinoma of the endometrium (CCC) is a poorly differentiated and aggressive subtype. They are characterized by an admixture of tubulocystic, papillary, and/or solid patterns and cuboidal, polygonal, hobnail, or flat cells with a clear eosinophilic cytoplasm. This type of cancer is often associated with worse clinical outcomes and is less responsive to hormone therapies, such as serous carcinoma. Clear cell carcinomas are rare but have a poor prognosis owing to their aggressive nature [2]. Undifferentiated carcinoma is defined by the absence of an apparent lineage of differentiation, with cells displaying extreme pleomorphism and high mitotic rates. These tumors are particularly aggressive and are often diagnosed at an advanced stage. There is a lack of typical structural organization, and these tumors may blend with more differentiated forms of endometrial carcinoma, creating diagnostic challenges [9]. Dedifferentiated carcinoma is a rare subtype that features both components of undifferentiated carcinoma and well-differentiated or moderately differentiated endometrioid carcinoma. They are considered highly aggressive, and their prognosis is poor because they are often associated with advanced disease at diagnosis [17]. Carcinosarcomas (malignant mixed Müllerian tumors) are classified separately from epithelial endometrial cancers; however, they contain both carcinomatous (epithelial) and sarcomatous (mesenchymal) components. The epithelial component is often poorly differentiated and includes features of serous or endometrioid carcinoma. Carcinosarcomas are clinically aggressive and have a poor prognosis owing to their high rate of metastasis [2].

PDEECs exhibit several distinct histomorphological features that distinguish them from their well-differentiated counterparts [18]. One of the hallmark features of PDEECs is pronounced nuclear atypia, which manifests as pleomorphic, hyperchromatic nuclei with irregular contours. This atypia can be extensive and is typically more pronounced than that in low-grade tumors [7]. PDEECs exhibit a high mitotic index, which reflects the rapid proliferation of tumor cells. This is a key indicator of aggressive behavior and correlates with the malignant potential of the tumor [11]. In contrast to well-differentiated endometrioid carcinomas, which display a well-defined glandular architecture, poorly differentiated tumors often lose this organization and adopt a solid or papillary growth pattern. This lack of glandular formation is a hallmark of high-grade ECs, particularly in subtypes such as serous carcinoma [2]. Well-differentiated ECs are characterized by clear gland formation. However, poorly differentiated variants often exhibit a near-total loss of glandular architecture, with disorganized and solid sheets of cells replacing gland formation [10].

Tumor necrosis and peritumoral inflammation are often present in PDEECs, further complicating histological evaluation and serving as additional indicators of tumor aggressiveness [19]. Another feature of PDEECs is the frequent invasion of lymphatic and blood vessels. This increases the likelihood of distant metastasis, contributing to the poor prognosis commonly associated with these cancers [11]. PDEECs may also exhibit features of serous carcinoma (e.g., papillary structures and eosinophilic cytoplasm) or clear cell carcinoma (e.g., clear or hobnail cells), which further complicates accurate histopathological classification [7]. These morphological markers help pathologists differentiate PDEECs from their well-differentiated counterparts but still present challenges due to the high degree of heterogeneity within these tumors [20].

One of the most significant challenges in diagnosing PDEECs is the variability of their histopathological features [21]. The overlapping characteristics of different high-grade tumors, such as high-grade endometrioid, serous, and undifferentiated carcinomas, can lead to substantial diagnostic uncertainty. Even among experienced pathologists, there can be considerable disagreement in the interpretation of PDEECs [7]. The lack of clear and consistent morphological boundaries between tumor subtypes complicates the task of providing a definitive diagnosis. For example, the solid growth patterns observed in poorly differentiated endometrioid carcinomas can resemble those found in serous or undifferentiated carcinomas [2]. The histological overlap between poorly differentiated EC subtypes can blur the distinction between these categories. This overlap in morphology can lead to misclassification, which can affect treatment decisions and ultimately affect patient outcomes [7].

Additionally, inter-observer variability among pathologists when assessing poorly differentiated tumors can lead to inconsistent diagnoses. Consequently, this traditional approach is often inadequate for predicting prognosis or informing tailored treatment strategies, particularly in cases of PDEECs [11]. The sole reliance on histomorphological evaluation often results in incomplete diagnostic information, especially for tumors that present ambiguous or mixed features. Thus, additional techniques, such as immunohistochemistry (IHC), have become crucial in achieving a more precise classification of these tumors [2]. Targeted IHC panels are employed to distinguish between specific high-grade endometrial cancer subtypes [22,23]. For example, p53, p16, HNF-1β, ER/PR, and mismatch repair (MMR) proteins are commonly used to differentiate serous carcinoma from clear cell carcinoma [23]. Likewise, markers such as ER, PR, napsin A, and AMACR aid in distinguishing endometrioid carcinoma from clear cell carcinoma [22].

However, despite their utility, IHC markers often lack the necessary sensitivity and specificity to provide definitive results [23]. This limitation arises because certain markers can yield overlapping expression patterns across different tumor types, leading to inconclusive findings [24]. For example, although p53 mutations are common in serous carcinoma, they can also appear in other high-grade endometrial cancers, reducing the reliability of p53 as a standalone marker. Similarly, napsin A and HNF-1β, although helpful in identifying clear cell carcinoma, may not provide absolute specificity owing to the variability in marker expression [22]. In practice, the limitations of IHC highlight the need to combine these markers with the clinical context and, potentially, other molecular techniques to improve diagnostic accuracy [23]. This has prompted a shift toward integrating molecular and genetic profiling into the diagnostic process, which offers a more precise understanding of tumor biology and allows for improved classification [8]. Misclassification of PDEECs can lead to inappropriate treatment strategies. For example, serous carcinoma often requires more aggressive treatment than high-grade endometrioid carcinoma, and incorrect diagnosis could lead to undertreatment or overtreatment, both of which have serious implications for patient prognosis [2].

In light of these challenges, the transition from traditional histopathological evaluation to molecular-based classification represents a critical advancement for improving the accuracy of diagnosis and treatment [2]. Molecular profiling provides a more robust framework for distinguishing poorly differentiated EC subtypes and predicting their clinical behavior. The evolution from morphology to molecular diagnostics holds promise for improving the precision of diagnosis, prognosis, and treatment strategies in challenging cases [25].

## 3. Molecular Classification of PDEECs

The molecular classification of PDEECs has revolutionized our understanding of these aggressive malignancies, providing insights into their pathogenesis and opening doors for targeted therapies [26]. Based on data from The Cancer Genome Atlas (TCGA), endometrial cancers, including poorly differentiated cases, are classified into four main molecular subtypes (Figure 3) [27]. The molecular subtyping provided by TCGA represents a shift from purely histological diagnosis to a more nuanced understanding of the genetic underpinnings of PDEECs, guiding personalized treatment and improving patient outcomes [28]. Molecular classification aligns with traditional histology in several ways but also reveals key discrepancies, particularly in poorly differentiated and high-grade tumors [29].

Most endometrioid ECs fall into the polymerase epsilon (POLE) ultramutated, MSI-Hypermutated, or Copy-Number Low (CNL) groups, reflecting their diverse molecular landscapes. However, high-grade endometrioid carcinomas can also fall into the Copy-Number High (CNH) category, which is typically associated with serous carcinomas, complicating the histological classification [26]. Serous carcinomas are aggressive tumors that align well with the CNH subgroup. Histologically, they exhibit significant nuclear atypia, high mitotic indices, and TP53 mutations, which are the defining features of this molecular group [16]. Historically considered a distinct histological entity, clear cell ECs have not been clearly segregated into a single molecular subgroup. They can be found across various TCGA molecular subtypes, underscoring the limitations of histology alone in predicting behavior and prognosis [30]. Undifferentiated PDEECs pose challenging cases, which often show heterogeneous molecular profiles, with some aligning with the CNH subgroup (TP53 mutations) and others falling into the MSI or NSMP categories [26]. Histologically, these tumors are difficult to classify based on morphology alone, and molecular profiling provides much-needed clarity [7]. Thus, while traditional histology remains useful, molecular classification often provides better discrimination between tumor types, especially in cases where histomorphology is ambiguous, such as PDEECs. Molecular profiling can help refine diagnostic accuracy and ensure appropriate treatment planning.

The prognostic implications of molecular classification are substantial, offering insights into survival outcomes, recurrence risk, and therapeutic responsiveness [11]. Molecular profiling provides superior prognostic information compared to traditional histopathological methods, particularly in poorly differentiated and high-grade tumors, where morphology may not reflect biological behavior [26]. Despite being histologically classified as high-grade, ultramutated POLE tumors have an excellent prognosis owing to their highly immunogenic nature. They are often associated with long-term survival and low recurrence rates, making them distinct from other high-grade ECs [31]. The moderate prognosis associated with MSI-hypermutated tumors highlights the potential for therapeutic interventions such as immune checkpoint inhibitors. MSI-H tumors exhibit higher response rates to PD-1/PD-L1 blockade due to the presence of neoantigens, enhancing the immunogenicity of the tumor [11].

CNL tumors, including most low-grade endometrioid carcinomas, generally have a favorable prognosis. These tumors often benefit from hormonal therapies, given their estrogen receptor positivity, and show lower recurrence rates than the CNH or MSI-H groups [26]. The poor prognosis of CNH tumors is largely due to their association with serous carcinomas and TP53 mutations. These tumors are aggressive, have a high likelihood of metastasis and recurrence, and are often resistant to standard therapies [32]. However, they may benefit from targeted therapies directed at specific molecular alterations (e.g., PI3K/AKT/mTOR inhibitors). In cases where histomorphology is ambiguous, molecular profiling provides critical prognostic insights [33]. For example, a poorly differentiated endometrioid carcinoma that falls into the CNH group (with TP53 mutations) would have a markedly different prognosis and treatment strategy compared to that in the POLE ultramutated or copy-number low groups [34]. This molecular differentiation allows for more tailored therapeutic approaches, reducing reliance on morphology alone, which can be misleading in poorly differentiated tumors [28]. Hence, the molecular classification of endometrial cancer, as established by TCGA, represents a significant advance in the diagnosis, prognosis, and treatment of ECs, especially poorly differentiated subtypes [7]. These four molecular subgroups not only refine the classification of EC but also provide valuable prognostic information. By correlating molecular profiles with histology and integrating this information into clinical practice, oncologists can better stratify patients, predict outcomes, and tailor treatments, ultimately improving the management of endometrial cancer [29].

## 4. Genomic Landscape of PDEECs

The genomic landscape of PDEECs reflects the complex molecular changes that drive the aggressive behavior and poor prognosis of these tumors [28]. The transition from histomorphology-based classification to a molecular-based understanding has uncovered several key genetic alterations, marked by distinct molecular subtypes with varying degrees of chromosomal instability, mutation burden, and pathway activation, as highlighted in Table 1 [26]. These genetic alterations provide insights into the mechanisms driving tumor development and offer potential avenues for targeted therapy. Deciphering the genomic landscape of PDEECs not only deepens the understanding of the biological underpinnings of these aggressive cancers but also opens the door for personalized therapeutic approaches [35].

### 4.1. TP53 Mutations

Mutations in TP53 play a pivotal role in the pathogenesis and progression of PDEECs, including more aggressive forms such as uterine serous carcinomas and high-grade endometrioid carcinomas [36]. TP53 mutations are strongly associated with aggressive tumor behavior, poor prognosis, and resistance to conventional therapies. The TP53 gene encodes the tumor suppressor protein p53, often referred to as the “guardian of the genome” due to its crucial role in maintaining genomic stability [37]. p53 functions by regulating cell cycle arrest, DNA repair, and senescence. It halts the cell cycle in response to DNA damage, allowing time for DNA repair [38]. p53 activates DNA repair mechanisms to ensure genomic integrity. If the damage is irreversible, p53 induces programmed cell death to prevent the propagation of defective cells. It also promotes permanent cell cycle arrest (senescence) in cells with irreparable damage [39]). Hence, under normal conditions, p53 functions as a key barrier to oncogenesis by preventing the accumulation of genetic mutations. Loss of p53 function through mutations can thus lead to uncontrolled cell proliferation and tumor development [11].

TP53 mutations are among the most frequent genetic alterations in PDEECs, particularly in aggressive subtypes. These mutations result in the loss of normal p53 function and the emergence of highly malignant phenotypes [7]. Most TP53 mutations in PDEECs are missense mutations, which lead to the production of dysfunctional p53. These mutations often occur in the DNA-binding domain of the protein, preventing it from effectively regulating the target genes involved in cell cycle control and apoptosis [40]. Additionally, nonsense mutations and deletions can result in truncated or completely absent p53. Uterine serous carcinoma (USC), an aggressive subtype of PDEECs, harbors TP53 mutations in over 90% of cases [37]. These mutations drive the highly proliferative, invasive, and treatment-resistant nature of these tumors. Unlike low-grade endometrioid carcinomas, which are often PTEN-driven and hormone-sensitive, serous carcinomas are characterized by p53 dysfunction and are typically estrogen receptor (ER) and progesterone receptor (PR) negative [41].

Loss of p53 function leads to genomic instability, a hallmark of PDEECs. Without functional p53 to halt cell division in the presence of DNA damage, cells accumulate further mutations that drive tumor progression and contribute to the heterogeneity of the tumor [36]. Genomic instability contributes to both intra- and inter-tumor heterogeneity, making treatment more challenging. Cells with TP53 mutations in PDEECs exhibit resistance to apoptosis [8]. For example, *TP53* mutations lead to the production of a mutant dysfunctional p53 protein that is significantly more stable and long-lived than wild-type (wt) p53, which is rapidly degraded by the ubiquitin–proteasome system [7,11]. As a result, while WT p53 typically generates a weak or transient signal in immunohistochemistry (IHC) owing to its short half-life, mutant p53 accumulates in tumor cells, producing a strong and persistent nuclear staining pattern [11]. This allows for the survival and proliferation of cancerous cells even in the face of chemotherapy or radiation treatment, which aims to induce apoptosis in malignant cells. Furthermore, research has demonstrated that mutant p53 not only loses its normal tumor suppressor function but also gains distinct molecular properties that contribute to cancer progression [36]. One key mechanism involves the ability of mutant p53 to form heterotetramers with the remaining wild-type p53, thereby interfering with its DNA-binding ability and transcriptional regulatory functions. This dominant-negative effect compromises the expression of critical p53 target genes involved in apoptosis, cell cycle arrest, and DNA repair, ultimately promoting tumorigenesis and resistance to therapy [41,42]. TP53 mutations in PDEECs are associated with poor prognosis and are characterized by a high recurrence rate, early metastasis, and shorter overall survival [43]. Unlike low-grade endometrial carcinomas, which often respond well to surgery and hormone therapy, PDEECs, especially those with TP53 mutations, tend to resist conventional therapies and require aggressive treatment approaches [36].

The presence of TP53 mutations in PDEECs poses significant therapeutic challenges but also opens avenues for targeted therapies. Several strategies are under investigation to restore or exploit the dysfunction of the p53 pathway in cancers with TP53 mutations [37]. This includes developing small molecules that can reactivate mutant p53 or stabilize wild-type p53 in tumors with partial functionality. One example is PRIMA-1 and APR-246, compounds designed to restore the normal function of mutated p53, thereby reactivating its tumor-suppressive properties [44]. Another promising approach is the concept of synthetic lethality, where two genetic defects together result in cell death, but either defect alone does not [45]. For TP53 mutations, researchers are exploring PARP inhibitors and other drugs that target the DNA repair pathways. Tumors with p53 deficiency rely on alternative DNA repair mechanisms, and inhibition of these pathways can selectively kill cancer cells while sparing normal cells [46]. Tumors with TP53 mutations often exhibit high levels of genomic instability and neoantigen load, potentially making them more responsive to immunotherapies, such as immune checkpoint inhibitors. Clinical trials to assess the effectiveness of immune-based therapies in treating TP53-mutant endometrial cancers have been undertaken [36].

Given the complex biology of PDEECs with TP53 mutations, combination therapies that simultaneously target multiple pathways may offer the best approach [47]. For example, combining PI3K/AKT/mTOR inhibitors with drugs targeting mutant p53 or DNA repair pathways could improve treatment outcomes [43]. Hence, TP53 mutations are a defining feature of PDEECs, particularly uterine serous carcinomas.

### 4.2. PTEN Mutations

PTEN (Phosphatase and Tensin Homolog) is a key tumor suppressor gene that plays a significant role in cellular processes, such as cell growth, survival, and proliferation. Mutations or loss of PTEN function are among the most common genetic alterations found in endometrial cancers, with up to 80%, especially in endometrioid subtypes [48,49]. However, the role of PTEN mutations in PDEECs is more nuanced, as these high-grade cancers exhibit distinct molecular profiles and more aggressive behavior than their lower-grade counterparts [50]. Under normal physiological conditions, PTEN functions primarily as a lipid phosphatase, regulating the PI3K/AKT/mTOR signaling pathway, which is critical for cell proliferation, survival, and metabolism [51]. PTEN negatively regulates this pathway by dephosphorylating PIP3 (phosphatidylinositol (3,4,5)-trisphosphate), converting it back to PIP2, thereby inhibiting AKT activation. This action prevents uncontrolled cellular proliferation and promotes apoptosis under conditions of cellular stress or damage [52]. Loss or inactivation of PTEN removes this critical regulatory checkpoint, leading to hyperactivation of the PI3K/AKT/mTOR pathway and promotion of tumorigenesis [7]. Thus, most PTEN mutations are inactivating mutations, leading to the loss of protein function and constitutive activation of the PI3K/AKT/mTOR pathway, which drives tumorigenesis.

PTEN mutations can be broadly categorized into missense mutations, frameshift/nonsense mutations, and deletions (Table 2), each with distinct molecular consequences. Missense mutations (hotspot mutations disrupting the active site) are single-nucleotide substitutions that alter a critical amino acid within the catalytic or regulatory domains of PTEN [51]. Most hotspot mutations occur in the phosphatase domain, leading to partial or complete loss of enzymatic activity. Some missense mutations retain a stable PTEN protein but disrupt its tumor-suppressor function [40]. The most common hotspot mutations include R130G/Q (Arg130Gln), C124S (Cys124Ser), H123Y (His123Tyr), and G129E (Gly129Glu). R130G/Q (Arg130Gln) directly disrupts the catalytic active site and abolishes phosphatase function. C124S (Cys124Ser) eliminates the nucleophilic cysteine residues required for enzymatic activity. H123Y (His123Tyr) impairs substrate binding and phosphatase activity, and G129E (Gly129Glu) affects core enzymatic function [28,50]. These mutations do not always result in the complete loss of the PTEN protein, but they prevent proper dephosphorylation of PIP3, leading to hyperactivation of PI3K/AKT signaling. Retained PTEN expression may exert tumor-suppressive effects through phosphatase-independent mechanisms [7].

Frameshift and nonsense mutations are truncating mutations that result in complete protein loss. They introduce premature stop codons, leading to the production of truncated, non-functional proteins or triggering nonsense-mediated decay (NMD), resulting in the complete loss of PTEN expression. These mutations occur through insertions, deletions (indels), or single nucleotide substitutions that disrupt the open reading frame [48,51]. Examples of common truncating mutations include R233 (Arg233Stop), Y68fs (Tyrosine68 frameshift), and R335 (Arg335Stop). R233 (Arg233Stop) produces a truncated protein lacking key functional domains, Y68fs (Tyrosine68 frameshift) alters protein translation, leading to premature degradation, and R335 (Arg335Stop) is a nonsense mutation eliminating the C-terminal regulatory domain [48,50,51,53]. These mutations are more aggressive than missense mutations because they lead to a total absence of PTEN protein rather than partial enzymatic impairment. Some cancers exhibit large deletions and gene copy loss encompassing the entire PTEN gene or significant portions of its coding region [7,50]. For example, loss of heterozygosity (LOH) at 10q23, the chromosomal locus of PTEN, is common in glioblastomas and PDEECs. Homozygous deletions lead to complete PTEN inactivation, promoting highly aggressive tumor phenotypes. Heterozygous deletions may cooperate with additional epigenetic silencing mechanisms to reduce PTEN activity [10,28,50].

PTEN mutations are most commonly associated with low-grade endometrioid ECs; however, they are also present in certain high-grade tumors or PDEECs, albeit at a lower frequency compared to that of TP53 mutations [28]. These mutations are particularly relevant in high-grade endometrioid carcinomas and mixed carcinomas, where poorly differentiated histological features are observed. PTEN mutations are less frequent in serous and clear cell endometrial cancers, which are subtypes of PDEECs [54]. In contrast, high-grade endometrioid carcinomas and mixed histology tumors may still harbor PTEN mutations [10]. The lower prevalence of PTEN mutations in serous carcinomas suggests that other genetic drivers, such as TP53 and PIK3CA mutations, may play more significant roles in these aggressive subtypes [36]. In PDEECs that retain PTEN mutations, the loss of PTEN function accelerates tumor progression by promoting unchecked cellular proliferation and resistance to apoptosis [28]. Hyperactivation of the PI3K/AKT/mTOR pathway leads to increased tumor growth and survival, contributing to the aggressive clinical behavior observed in these cancers [10]. The loss of PTEN, in conjunction with other genetic alterations, such as PIK3CA or TP53 mutations, creates a synergistic effect that drives the malignant potential of PDEECs [11].

PTEN loss has been associated with dedifferentiation in endometrial cancers, indicating that tumors with PTEN mutations may transition from low-grade, well-differentiated forms to poorly differentiated or high-grade variants [55]. Dedifferentiated endometrial carcinomas, which show histological features of both well-differentiated and undifferentiated components, often have underlying PTEN mutations that contribute to their progression from a more indolent to a more aggressive phenotype. This dedifferentiation process may be linked to genomic instability and alterations in other signaling pathways [56]. PTEN loss often occurs alongside mutations in the PIK3CA gene, which encodes the catalytic subunit of PI3K, further enhancing the PI3K/AKT/mTOR signaling pathway. The co-occurrence of PTEN and PIK3CA mutations creates a dual hit on the regulatory mechanisms controlling cell growth and survival, leading to increased oncogenic potential [7]. Additionally, in PDEECs, where both PTEN loss and TP53 mutations are present, the tumor suppressive machinery is severely compromised, leading to aggressive tumor behavior, resistance to apoptosis, and poor prognosis [36].

PTEN mutations in endometrial cancer are generally associated with a favorable prognosis in lower-grade tumors; however, in the context of PDEECs, particularly those that have progressed from lower-grade endometrioid forms, loss of PTEN function is indicative of a more aggressive disease course [57]. In these high-grade tumors, PTEN loss contributes to rapid tumor growth, early metastasis, and therapeutic resistance, particularly in therapies targeting the PI3K/AKT/mTOR pathway [10]. The identification of PTEN mutations in PDEECs offers potential opportunities for targeted therapies, although the complex molecular landscape of these tumors poses challenges for effective treatment [28]. Given the role of PTEN in negatively regulating the PI3K/AKT/mTOR pathway, inhibitors targeting this pathway have shown promise in preclinical studies and early-phase clinical trials for endometrial cancer [58]. These inhibitors aim to block the downstream effects of PTEN loss, potentially slowing tumor progression and enhancing the effectiveness of conventional therapies. However, resistance to single-agent inhibitors is common, and combination therapies with other agents targeting different pathways (e.g., MEK inhibitors or immune checkpoint inhibitors) are being explored [59].

The absence of functional PTEN creates specific vulnerabilities in cancer cells. One approach is the use of synthetic lethality, where therapies target secondary pathways that are essential for survival in PTEN-deficient tumors. For example, PARP inhibitors are being explored in PTEN-deficient tumors owing to their reliance on alternative DNA repair pathways [60]. The role of PTEN in modulating the tumor immune microenvironment is an emerging area of research. Loss of PTEN has been associated with an immunosuppressive tumor microenvironment, potentially influencing the response to immune checkpoint inhibitors [56]. Ongoing trials are assessing the combination of immunotherapy and PI3K/AKT/mTOR pathway inhibitors in PTEN-mutant endometrial cancers, including poorly differentiated EC subtypes. Given the complexity of genetic alterations in PDEECs, combination therapies that simultaneously target multiple pathways may offer the best therapeutic approach [61]. Combinations of PI3K inhibitors with agents targeting the RAS/RAF/MEK or Wnt signaling pathways are being tested to overcome the resistance mechanisms that arise from single-pathway inhibition [62]. Thus, PTEN mutations play a significant role in the pathogenesis of certain PDEECs, particularly high-grade endometrioid carcinomas and mixed histologies. Although PTEN mutations are less frequent in serous carcinomas than TP53 mutations, they still represent an important molecular alteration in the broader spectrum of PDEEC.

### 4.3. PIK3CA Mutations

PIK3CA, which encodes the catalytic subunit p110α of phosphoinositide 3-kinase (PI3K), is one of the most frequently mutated genes in various cancers, including endometrial cancer. Mutations in PIK3CA are particularly relevant in the development and progression of PDEECs, which are high-grade and aggressive subtypes of endometrial carcinomas [63]. These cancers, including serous carcinomas, clear cell carcinomas, and some high-grade endometrioid tumors, are molecularly distinct from their well-differentiated counterparts [64]. PIK3CA plays a crucial role in regulating the PI3K/AKT/mTOR pathway, which controls critical cellular functions such as cell growth, survival, metabolism, and proliferation [65]. Under normal conditions, PI3K is activated by receptor tyrosine kinases (RTKs) and G-protein coupled receptors (GPCRs), leading to the phosphorylation of PIP2 (phosphatidylinositol 4,5-bisphosphate) into PIP3 (phosphatidylinositol 3,4,5-triphosphate) [66]. PIP3 recruits and activates AKT, leading to the activation of downstream signaling pathways, such as mTOR. This pathway is critical for maintaining cellular homeostasis, and its dysregulation often results in unchecked cellular growth, survival, and cancer development [67].

Mutations in PIK3CA are typically activating mutations, which enhance PI3K signaling and occur in about 20–30% of endometrial cancers. While they account for about 40–50% of endometrioid subtypes, they also appear in poorly differentiated subtypes, such as serous and clear cell carcinomas, although at a lower frequency. Importantly, these mutations contribute significantly to the aggressive nature and therapeutic resistance of PDEECs [68]. PIK3CA mutations in PDEECs are classified into two main categories: hotspot missense mutations (gain-of-function) and other rare mutations. The most common missense mutations in PIK3CA are located in two hotspot regions: exon 9 and exon 20. Exon 9 affects the helical domain of p110α, while exon 20 affects the kinase domain of p110α. E542K mutation in exon 9 disrupts p85-mediated inhibition, leading to constitutive PI3K activation. Similar to E542K, the E545K mutation results in the loss of negative regulation by p85. Helical domain mutations (E542K and E545K) relieve inhibitory interactions between p85 (regulatory subunit) and p110α, making PI3K more active even in the absence of upstream growth factor stimulation. H1047R mutations occur in exon 20 and directly enhance catalytic activity, increasing downstream signaling. Mutations in the H1047R kinase domain increase substrate affinity and catalytic efficiency, leading to sustained activation of PI3K/AKT/mTOR signaling [67]. Thus, the mutations lead to constitutive activation of the PI3K pathway, independent of external growth signals, thus promoting cell proliferation, survival, and metastasis [65].

Other rare PIK3CA mutations in PDEECs include in-frame insertions or deletions and copy number amplifications. In-frame insertions or deletions occur within the kinase domain and can enhance enzymatic function similarly to H1047R, whereas copy number amplifications of PIK3CA may also contribute to pathway hyperactivation in some PDEECs [64]. Mutant PIK3CA results in a constitutively active PI3K enzyme, leading to uncontrolled activation of downstream signaling, even in the absence of growth factor stimulation. The mechanism involves loss of inhibition by p85 (regulatory subunit), increased lipid kinase activity, and activation of downstream targets [63,65]. p85 subunit helical domain mutations (E542K, E545K) prevent binding to PIK3CA, leading to unrestrained PI3K activity and elevated AKT signaling leading to enhanced cancer cell proliferation and viability [67]. In the context of increased lipid kinase activity, the mutant p110α catalytic subunit converts PIP2 (phosphatidylinositol-4,5-bisphosphate) into PIP3 (phosphatidylinositol-3,4,5-trisphosphate) at a higher rate than wild-type PI3K [68]. In the activation of downstream targets, increased PIP3 levels at the plasma membrane recruit and activate AKT (protein kinase B) via phosphoinositide-dependent kinase-1 (PDK1). Activated AKT then phosphorylates and regulates multiple downstream effectors that drive pro-tumorigenic processes [64,65,66,67]. Once activated, mutant PI3K initiates a cascade of signaling events that contribute to the aggressive behavior of PDEECs, as highlighted in Table 3 [66].

PIK3CA mutations interact with other genetic alterations in PDEECs and rarely occur in isolation. Mutations often co-exist with alterations in other critical genes, creating a complex molecular landscape that drives aggressive tumor behavior [69]. PIK3CA mutations often co-occur with TP53 mutations in serous carcinomas and other high-grade endometrial cancers. The loss of TP53 function (a key tumor suppressor involved in DNA repair and apoptosis), along with the activation of PI3K signaling, creates a powerful oncogenic environment that enhances genomic instability and tumor progression [70]. Loss of PTEN is frequently observed alongside PIK3CA mutations in endometrioid endometrial cancers [7]. In PDEECs, PTEN loss further amplifies the oncogenic effects of PIK3CA mutations, contributing to the aggressiveness of these tumors. In certain high-grade endometrial cancers, mutations in ARID1A, a chromatin remodeling gene, and RAS pathway genes frequently co-occur with PIK3CA mutations [63]. These genetic alterations collectively drive tumor progression, treatment resistance, and poor clinical outcomes. Given the central role of PIK3CA mutations in driving the pathogenesis of PDEECs, this gene and its associated signaling pathway have become key therapeutic targets [67].

According to ref. [71], PIK3CA mutations play a critical role in the pathogenesis of PDEECs, driving oncogenic processes through activation of the PI3K/AKT/mTOR pathway [72]. According to ref. [63], targeting the PI3K pathway represents a promising approach to improve outcomes in patients with difficult-to-treat cancers. However, the heterogeneity of PDEECs suggests that personalized, multi-targeted therapeutic strategies are necessary to achieve optimal results [73].

### 4.4. ARID1A Mutations:

ARID1A (AT-rich interaction domain 1A) is a tumor-suppressor gene encoding a key component of the SWI/SNF chromatin remodeling complex, which plays a crucial role in regulating gene expression through the modification of chromatin structure. Mutations in ARID1A are frequently observed in several types of cancer, including PDEECs [74]. These cancers include aggressive subtypes, such as high-grade endometrioid carcinomas, serous carcinomas, and clear cell carcinomas, where ARID1A mutations contribute significantly to tumor progression, genomic instability, and treatment resistance [75]. The SWI/SNF complex is responsible for altering the structure of chromatin, allowing or restricting access to transcriptional machinery, thus regulating gene expression [76]. ARID1A, as a part of this complex, promotes the expression of genes involved in various cellular processes, including differentiation, cell cycle regulation, and DNA damage repair [63]. Loss of ARID1A function due to mutations leads to impaired chromatin remodeling and dysregulation of transcriptional programs that are critical for maintaining genomic integrity [77]. In endometrial cancer, ARID1A mutations result in epigenetic alterations that drive oncogenesis by disrupting normal cell growth, differentiation, and DNA repair mechanisms [67].

Mutations in ARID1A are more commonly associated with endometrioid and clear cell subtypes of endometrial carcinoma, with mutation rates reported between 40 and 50% in endometrioid tumors and 30–50% in clear cell carcinomas [63]. However, poorly differentiated endometrial carcinomas, including high-grade endometrioid carcinomas, may also harbor ARID1A mutations [67]. These mutations tend to co-occur with other genetic alterations that contribute to disease aggressiveness. Loss-of-function mutations in ARID1A typically result in the inactivation of the SWI/SNF complex, leading to several oncogenic consequences in PDEECs [78]. ARID1A is involved in the regulation of genes that control the cell cycle, such as p21 and p53. Loss of ARID1A function can lead to uncontrolled cell proliferation due to failure to induce cell cycle arrest in response to DNA damage or oncogenic stress [67]. Loss of ARID1A impairs the cell’s ability to maintain chromatin structure and properly repair DNA damage [79]. This leads to increased genomic instability, which promotes the accumulation of additional genetic alterations that drive tumor progression. In PDEECs, this results in rapid tumor growth and metastasis [65].

In some endometrial cancers, ARID1A loss has been linked to the activation of oncogenic pathways, such as the PI3K/AKT/mTOR signaling pathway [80]. This relationship is significant because it suggests that ARID1A mutations may cooperate with other genetic mutations, such as PIK3CA or PTEN loss, to promote aggressive tumor behavior [74]. Loss of chromatin remodeling due to ARID1A mutations can lead to widespread epigenetic changes, including aberrant gene silencing and activation of oncogenes. This contributes to tumor heterogeneity and resistance to treatment, particularly in PDEECs. ARID1A mutations often occur alongside other key genetic alterations in PDEECs, exacerbating the aggressive nature of the disease [78]. ARID1A mutations are frequently found in conjunction with mutations in the PI3K pathway, particularly in PIK3CA and PTEN. The loss of ARID1A in these contexts may enhance oncogenic signaling through the PI3K/AKT/mTOR pathway, contributing to unchecked cell growth and survival in poorly differentiated tumors [81]. The co-occurrence of ARID1A and TP53 mutations is common in serous endometrial carcinomas, a high-grade subtype of PDEECs. TP53 mutations disrupt the normal function of p53, a crucial tumor suppressor involved in apoptosis and DNA repair, leading to aggressive tumor behavior and poor clinical outcomes when combined with ARID1A loss [63].

Emerging research suggests that ARID1A mutations may also influence the tumor microenvironment and immune response in endometrial cancers [77]. Loss of ARID1A has been associated with altered expression of immune-related genes, potentially creating an immunosuppressive microenvironment that allows tumors to evade immune detection [82]. This aspect of ARID1A function could have significant implications in the development of immunotherapies targeting PDEECs with ARID1A mutations. Understanding the role of ARID1A mutations in PDEECs opens the door to targeted therapeutic approaches [77]. Several strategies have been explored to target ARID1A-mutant tumors. One promising approach involves exploiting the concept of synthetic lethality, where the loss of ARID1A renders tumors more vulnerable to the inhibition of other pathways involved in DNA damage repair [83]. For example, PARP inhibitors have shown potential for treating ARID1A-deficient tumors by targeting defective DNA repair mechanisms [84]. Given the role of ARID1A in chromatin remodeling, drugs that target epigenetic regulators, such as histone deacetylase (HDAC) inhibitors or DNA methyltransferase inhibitors, may offer therapeutic benefits to patients with ARID1A-mutant cancers [7]. The potential role of ARID1A in modulating the immune microenvironment suggests that immune checkpoint inhibitors may be effective in treating ARID1A-mutant PDEECs. Ongoing research is evaluating the impact of ARID1A mutations on the efficacy of immunotherapies, such as anti-PD-1/PD-L1 inhibitors [63].

The presence of ARID1A mutations in PDEECs is generally associated with a more aggressive disease course, higher-grade tumors, and poorer clinical outcomes [77]. Tumors harboring ARID1A mutations tend to exhibit increased genomic instability and resistance to conventional therapies, making them more challenging to treat. However, the precise prognostic significance of ARID1A mutations in PDEECs may vary depending on the tumor subtype and the presence of coexisting mutations [63]. In some cases, ARID1A mutations have been linked to improved responses to PARP inhibitors or other targeted therapies, suggesting that these mutations may have predictive value for certain treatments [85]. Thus, ARID1A mutations play a pivotal role in the pathogenesis of PDEECs, contributing to tumor development through the disruption of chromatin remodeling, genomic instability, and aberrant cell cycle regulation [77]. Loss of ARID1A function is often accompanied by other genetic alterations, such as mutations in PIK3CA, PTEN, and TP53, further driving the aggressive behavior of these cancers [80]. According to ref. [77], as research on ARID1A and its role in endometrial cancer progresses, new insights into the molecular underpinnings of these tumors may pave the way for more effective treatments and improved patient outcomes.

### 4.5. CTNNB1 Mutations

Mutations in CTNNB1, which encodes β-catenin, have increasingly been recognized as significant drivers of various cancers, including PDEECs [86]. These mutations play a crucial role in tumorigenesis by promoting uncontrolled cell growth, disrupting cellular adhesion, and facilitating tumor progression through aberrant activation of the Wnt/β-catenin signaling pathway [87]. In PDEECs, particularly high-grade endometrioid adenocarcinomas, CTNNB1 mutations contribute to more aggressive cancer phenotypes and poorer clinical outcomes [88]. CTNNB1 encodes β-catenin, a multifunctional protein that plays a dual role in cells. β-catenin is a key component of the cadherin complex, which is critical for maintaining cell-to-cell adhesion, thereby helping to preserve tissue architecture. β-catenin is also a central player in the Wnt signaling pathway, which regulates cellular proliferation, differentiation, and survival [89]. Under normal conditions, this pathway is tightly regulated, with β-catenin targeted for degradation in the absence of Wnt signaling. However, mutations in CTNNB1 disrupt these normal regulatory processes, leading to aberrant activation of the Wnt/β-catenin pathway and contributing to cancer development [90]. CTNNB1 mutations are most commonly associated with endometrioid endometrial carcinomas (EECs). These mutations are found in approximately 20–40% of low-grade EECs but are also present, albeit at lower frequencies, in poorly differentiated or high-grade forms of the disease [25]. In particular, high-grade endometrioid carcinomas with CTNNB1 mutations display aggressive behavior, with poorer prognosis and an increased likelihood of recurrence compared to those without the mutation [91].

Mutations in CTNNB1 often result in the stabilization of β-catenin in the cytoplasm and its subsequent translocation into the nucleus, where it acts as a transcriptional co-activator of Wnt target genes [25]. Normally, in the absence of Wnt signaling, β-catenin binds to a destruction complex, which includes APC, AXIN1, and GSK3-β, and is subsequently degraded [92]. However, mutations in the N-terminal phosphorylation sites of CTNNB1 prevent its degradation, allowing β-catenin to accumulate in the nucleus and activate Wnt target genes, leading to oncogenic processes such as uncontrolled cell proliferation and disruption of cellular adhesion [93]. Wnt target genes include MYC and CCND1 (which encodes Cyclin D1), both of which promote cell cycle progression and proliferation [90]. As a component of the cadherin complex, β-catenin mutations can weaken cell–cell adhesion, facilitating tumor invasion and metastasis. This is particularly significant in the context of poorly differentiated cancers, which are characterized by the loss of normal tissue architecture [94].

In PDEECs, CTNNB1 mutations have several specific oncogenic effects that contribute to their aggressive behavior [25]. In high-grade tumors, CTNNB1 mutations disrupt cellular adhesion mechanisms, allowing cancer cells to break away from the primary tumor mass, invade surrounding tissues, and potentially spread to distant organs [95]. This is a hallmark of PDEECs, which often exhibit aggressive clinical behavior and are associated with poor prognosis. Aberrant activation of the Wnt/β-catenin pathway through CTNNB1 mutations not only promotes cell proliferation but also enhances survival mechanisms. This makes tumor cells more resistant to apoptotic signals, contributing to treatment resistance, particularly to chemotherapy and radiation therapy [25]. Mutations in CTNNB1 may also promote cancer stem cell characteristics that are thought to be responsible for tumor recurrence and metastasis. These stem-like cells can evade traditional therapies and reinitiate tumor growth after treatment [4].

The impact of CTNNB1 mutations is often modulated by other genetic alterations. For example, in endometrial cancers, mutations in CTNNB1 frequently co-occur with mutations in PIK3CA, PTEN, and ARID1A, particularly in high-grade endometrioid carcinomas [96]. The combination of these mutations can result in synergistic oncogenic effects, driving the rapid progression of the disease. Loss of PTEN, a tumor suppressor gene, results in activation of the PI3K/AKT signaling pathway, promoting cell survival and proliferation [97]. When coupled with CTNNB1 mutations, which activate the Wnt/β-catenin pathway, dual activation of these pathways can lead to enhanced tumor growth and resistance to treatment [94]. Mutations in PIK3CA, which encodes a subunit of the PI3K enzyme, are also frequently found alongside CTNNB1 mutations in endometrial cancer. This co-occurrence can further drive oncogenic signaling and contribute to the poor prognosis observed in some cases of PDEECs [25]. The presence of CTNNB1 mutations in PDEECs is generally associated with more aggressive clinical behavior. These mutations have been linked to early tumor recurrence, resistance to standard therapies, and poorer overall survival outcomes [98]. In high-grade endometrial cancers, loss of cell adhesion and activation of oncogenic signaling pathways driven by CTNNB1 mutations contribute to rapid disease progression and metastasis [28]. However, the prognostic significance of CTNNB1 mutations can vary depending on the tumor subtype and other molecular features. For instance, while low-grade endometrioid carcinomas with CTNNB1 mutations may exhibit relatively indolent behavior, high-grade tumors tend to have a more aggressive clinical course [96].

Given the role of CTNNB1 mutations in activating the Wnt/β-catenin signaling pathway, targeting this pathway has emerged as a potential therapeutic strategy for endometrial cancers with CTNNB1 mutations [28]. While no Wnt pathway inhibitors are currently approved for clinical use in endometrial cancer, several small-molecule inhibitors targeting various components of the pathway, including β-catenin, are under investigation in preclinical and early clinical trials [25]. Additionally, the discovery of synthetic lethality involving Wnt signaling suggests that CTNNB1-mutant cancers may be vulnerable to combination therapies targeting other pathways that intersect with Wnt signaling, such as the PI3K/AKT/mTOR pathway [99]. Thus, CTNNB1 mutations play a critical role in the pathogenesis of PDEECs, particularly through aberrant activation of the Wnt/β-catenin signaling pathway [96]. These mutations contribute to aggressive tumor growth, invasion, and resistance to therapy, making them important molecular features in the progression of high-grade endometrial cancers [28]. As research continues to uncover the full impact of CTNNB1 mutations on endometrial cancer biology, they may also emerge as valuable targets for novel therapeutic strategies aimed at improving the outcomes of patients with these aggressive tumors.

### 4.6. POLE Mutations

POLE, encoding DNA polymerase epsilon, plays a crucial role in DNA replication and repair by providing high-fidelity synthesis through its exonuclease (proofreading) domain [100]. Mutations in this exonuclease domain are increasingly recognized for their impact on endometrial cancer, particularly in PDEECs, where they contribute significantly to the genetic and biological behavior of the cancer [101]. POLE, located on chromosome 12q24, encodes the catalytic subunit of DNA polymerase epsilon, which is primarily responsible for leading-strand synthesis during DNA replication [102]. The exonuclease domain of POLE acts as a proofreading mechanism, recognizing and excising incorrect nucleotides during DNA synthesis to maintain high fidelity [100]. Mutations in this domain impair the proofreading capability of the enzyme, resulting in an elevated error rate during replication [101]. POLE exonuclease domain mutations (shown in Figure 4) are observed in a subset of PDEECs, although their presence is generally more common in certain high-grade, aggressive subtypes rather than across all PDEECs [31]. These mutations are especially prevalent in tumors exhibiting an “ultramutated” phenotype, defined by an exceptionally high tumor mutation burden (TMB), which is due to the defective DNA proofreading function of the POLE exonuclease domain [103]. They also exhibit an increased CD8+ T cell presence and PD-L1 expression, making them highly responsive to immune checkpoint inhibitors. Moreover, hotspot mutations, such as POLE p.P286R, p.S297F, and p.V411L, result in defective exonuclease activity, increasing the accumulation of somatic mutations [34,100]. The accumulation of POLE exonuclease domain mutations contributes to genomic instability, which can drive tumorigenesis by enabling the rapid acquisition of oncogenic mutations. This process may support the aggressive and poorly differentiated nature of certain PDEECs, allowing cancer cells to adapt to and evade therapeutic pressures [101].

POLE-mutant PDEECs paradoxically exhibit a better prognosis despite their high mutation load [104]. Studies suggest that POLE-mutant tumors are more immunogenic owing to their high TMB, resulting in increased immune infiltration within the tumor microenvironment, which enhances antitumor immune responses [101]. POLE-mutated tumors represent a unique molecular subgroup in endometrial cancer, often associated with improved outcomes, even among cases with poor differentiation. This favorable prognosis differentiates them from other high-risk endometrial cancer groups [31]. POLE mutation status serves as an important biomarker for prognosis, guiding risk stratification, and treatment decisions in endometrial cancers. Knowing a patient’s POLE status can help identify those who may have favorable outcomes despite other high-risk clinical features, such as advanced stage or poor differentiation [101]. High TMB in POLE-mutant cancers leads to the generation of numerous neoantigens, which are more likely to be recognized by the immune system. This elevated neoantigen load makes POLE-mutant EECs particularly responsive to immune checkpoint inhibitors (e.g., anti-PD-1/PD-L1 therapies) because immune cells can better recognize and attack cancer cells [100]. The immunogenic potential of POLE-mutant tumors often results in a robust immune response that can reduce the biological aggressiveness of these cancers. As a result, patients with POLE-mutant PDEECs may achieve better responses to immunotherapy than those without POLE mutations, thus reshaping therapeutic approaches [31].

**Figure 4 cells-14-00382-f004:**
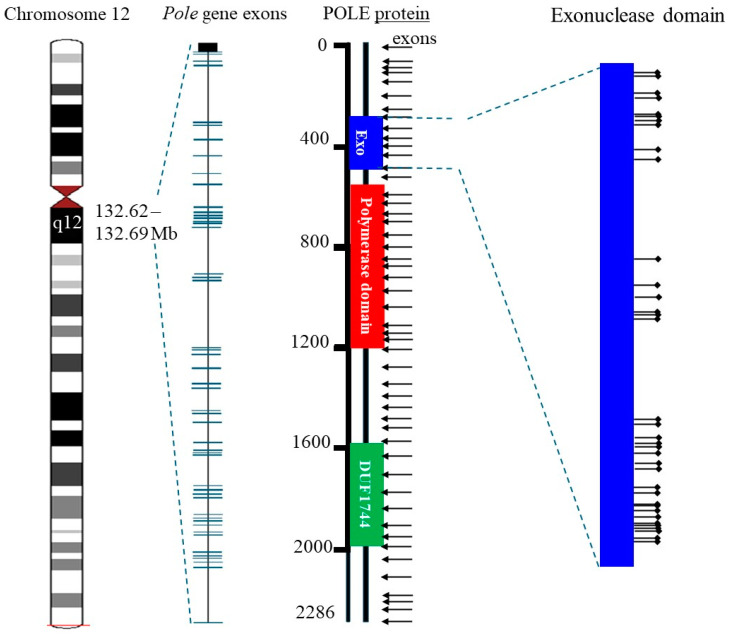
POLE mutations within the exonuclease domain. POLE exonuclease domain mutations significantly influence PDEEC biology. They drive a hypermutated phenotype that promotes immune recognition, often leading to better patient outcomes despite high tumor grades. This unique profile positions POLE as a valuable prognostic and predictive marker in PDEECs, guiding therapeutic approaches that prioritize immune-based treatments (figure adapted from [105]).

Beyond immunotherapy, POLE mutations also indicate that traditional chemotherapy may be less critical for achieving optimal outcomes in this subset of endometrial cancers [106]. Instead, clinical management should focus on immunotherapeutic strategies, especially given the favorable prognosis associated with these mutations [107]. Ongoing research is investigating the potential of targeting specific pathways associated with POLE mutations, such as enhancing the immune response through additional checkpoint inhibitors or combining immunotherapy with other agents that further boost immune recognition [101]. The increasing recognition of POLE as a biomarker has led to calls for routine testing of endometrial cancers. Assessing the POLE mutation status in patients with PDEECs can support personalized and effective treatment planning [4]. Studies continue to explore how POLE mutations interact with other genomic alterations, such as those in the PI3K/AKT/mTOR pathway or p53 signaling [108]. Understanding these interactions could yield insights into combination therapies and strategies to overcome potential resistance mechanisms in poorly differentiated cancers [109]. Thus, POLE exonuclease domain mutations significantly influence PDEEC biology. They drive a hypermutated phenotype that promotes immune recognition, often leading to better patient outcomes despite high tumor grades [100]. This unique profile positions POLE as a valuable prognostic and predictive marker in PDEECs, thereby guiding therapeutic approaches that prioritize immune-based treatments [108]. Ongoing research aims to further elucidate the mechanisms underpinning POLE-related tumorigenesis and identify optimal therapeutic strategies to exploit the heightened immunogenicity of POLE-mutant endometrial cancers [101]

### 4.7. Copy Number Alterations

Copy number alterations (CNAs), also referred to as copy number variations (CNVs), are changes in the number of copies of particular genes or genomic regions, involving either duplications (gains) or deletions (losses) of large DNA segments [110]. In PDEECs, CNAs play a significant role in tumor development, progression, and treatment resistance. These alterations contribute to genomic instability, influencing gene expression, signaling pathways, and overall tumor behavior [111]. PDEECs are often characterized by high genomic instability, which is a hallmark of aggressive and treatment-resistant cancers. This genomic instability manifests as large-scale CNAs that affect key oncogenes and tumor suppressor genes [110]. In PDEECs, CNAs are commonly observed across several regions of the genome, leading to aberrations in the regulation of cell growth, apoptosis, DNA repair, and other critical cellular functions. The most common CNAs identified in endometrial cancers (including poorly differentiated subtypes) involve amplification of oncogenes and deletion of tumor suppressor genes [13]. These alterations contribute to tumor progression and are often correlated with worse clinical outcomes. Several studies have identified common regions of copy number gain in PDEECs that are associated with oncogene overexpression [110].

Amplifications in the chromosome 1q region are frequently associated with increased expression of MYC, a well-known oncogene. MYC overexpression drives cell proliferation and is linked to aggressive cancer phenotypes and poor differentiation in endometrial tumors [112]. The chromosome 3q region often shows amplifications involving PIK3CA, an oncogene that encodes the catalytic subunit of PI3K. Overexpression of PIK3CA contributes to the activation of the PI3K/AKT signaling pathway, promoting cell growth, survival, and resistance to apoptosis, particularly in high-grade and poorly differentiated tumors [110]. Chromosome 8q is another key region involved in PDEECs, and amplification is often linked to c-MYC overexpression. The oncogenic properties of c-MYC lead to enhanced tumor proliferation and invasiveness, thereby contributing to the aggressive nature of PDEECs [113]. The chromosome 17q region is frequently associated with ERBB2/HER2 amplifications, particularly in serous endometrial cancers (a high-grade subtype often associated with poor differentiation). Overexpression of HER2, a receptor tyrosine kinase, drives oncogenic signaling and is linked to aggressive tumor growth and poor prognosis [114].

Conversely, regions of copy number loss often harbor tumor suppressor genes that, when deleted, contribute to tumor progression by removing constraints on cell proliferation and survival [115]. The key regions of copy number loss in PDEECs include chromosome 10q, chromosome 17p, and chromosome 18q. Loss of the chromosome 10q region is frequently associated with the PTEN tumor suppressor gene, one of the most commonly altered genes in endometrial cancers [110]. Loss of PTEN leads to dysregulation of the PI3K/AKT pathway, promoting unchecked cell proliferation and survival. Loss of PTEN is a hallmark of aggressive endometrial cancers, contributing to poor differentiation and worse clinical outcomes [55]. The chromosome 17p region includes TP53, a critical tumor suppressor frequently mutated in poorly differentiated and high-grade endometrial cancers [116]. The loss of TP53, whether through mutation or copy number loss, leads to genomic instability, loss of apoptotic control, and increased tumor aggressiveness [115]. Deletions in the chromosome 18q region are often linked to the loss of SMAD4, a tumor suppressor involved in TGF-β signaling. Loss of SMAD4 contributes to tumor progression by impairing cell differentiation and promoting epithelial–mesenchymal transition (EMT), a process associated with increased invasiveness and metastasis [110].

PDEECs, particularly high-grade endometrioid and serous subtypes, exhibit more complex patterns of CNAs than lower-grade or well-differentiated tumors. This complexity reflects greater genomic instability in these aggressive forms of cancer [110]. For example, endometrioid endometrial cancers typically show a spectrum of CNAs, with low-grade tumors having fewer alterations, whereas high-grade endometrioid carcinomas exhibit extensive CNAs affecting both oncogenes and tumor suppressors [117]. Serous endometrial carcinomas, another poorly differentiated subtype, are characterized by widespread CNAs, with amplifications of ERBB2 and MYC, and deletions of TP53 and RB1. These alterations are closely linked to the highly aggressive nature of serous cancers and their poor prognosis [118]. The presence of specific CNAs has significant clinical implications in patients with PDEECs. Certain CNAs, such as amplifications of ERBB2 or MYC or deletions of TP53 and PTEN, are associated with poor prognosis, higher rates of recurrence, and decreased overall survival [119]. Identifying these alterations can help stratify patients based on their risk of disease progression. CNAs can also influence therapeutic decisions. For instance, endometrial cancers with HER2 amplification may respond to HER2-targeted therapies (e.g., trastuzumab), similar to breast cancer treatment [110].

Additionally, tumors with PIK3CA amplifications or PTEN loss may be more susceptible to PI3K/AKT/mTOR pathway inhibitors. Certain CNAs may confer resistance to conventional therapies, such as chemotherapy or radiation. For example, loss of TP53 is often associated with resistance to DNA-damaging agents due to impaired apoptotic responses [120]. Understanding CNA profiles can guide the use of alternative therapeutic strategies, such as targeted therapies or immune checkpoint inhibitors. Advances in genomic technologies, including next-generation sequencing (NGS) and array-based comparative genomic hybridization (aCGH), have facilitated the identification of CNAs in PDEECs [121]. Thus, CNAs play a critical role in the development and progression of PDEECs by influencing tumor behavior, prognosis, and treatment responses [119]. These alterations often affect key oncogenes and tumor suppressor genes, leading to dysregulation of signaling pathways that drive aggressive tumor growth and resistance to therapy [110]. Understanding the CNA landscape in these cancers can provide valuable insights into prognostic indicators and potential therapeutic targets, paving the way for more personalized and effective treatment strategies for patients with high-risk tumors [121].

### 4.8. Epigenetic Modifications

Epigenetic modifications are heritable changes in gene expression that do not involve alterations to the underlying DNA sequence. These changes play a crucial role in cancer biology by influencing the behavior of cancer cells, including proliferation, survival, and metastasis [122]. In PDEECs, epigenetic modifications significantly contribute to tumor progression, aggressive clinical behavior, and resistance to therapy. The key epigenetic mechanisms involved in PDEECs include DNA methylation, histone modification, and non-coding RNA regulation [123]. DNA methylation is one of the most studied epigenetic alterations in cancer and involves the addition of a methyl group to cytosine bases in CpG islands, which are often found in gene promoter regions. Methylation typically leads to gene silencing, particularly in tumor suppressor genes [25]. In PDEECs, hypermethylation of CpG islands in the promoter regions of tumor suppressor genes is a common mechanism of gene inactivation. This can result in the loss of function of crucial regulators of cell growth and apoptosis, contributing to uncontrolled proliferation and tumor progression [124]. For instance, hypermethylation of the PTEN promoter, a key tumor suppressor in the PI3K/AKT pathway, is frequently observed in high-grade endometrial cancers. This leads to reduced expression of PTEN, enhanced PI3K signaling, and promotion of aggressive cancer phenotypes [69].

Hypermethylation of MLH1, a DNA mismatch repair gene, is often associated with microsatellite instability (MSI), a hallmark of certain types of endometrial cancers, including poorly differentiated subtypes [124]. Loss of MLH1 expression due to promoter methylation leads to defects in DNA repair, contributing to genomic instability and cancer progression. Although promoter hypermethylation silences tumor suppressor genes, global hypomethylation is also observed in PDEECs [125]. Hypomethylation of repetitive DNA sequences can lead to chromosomal instability, which is a key feature of aggressive and poorly differentiated tumors. This instability promotes oncogene activation, further driving tumor growth and metastasis [123,126].

Histone modifications are post-translational changes in histone proteins, around which DNA is wound. These modifications, including acetylation, methylation, phosphorylation, and ubiquitination, can influence the chromatin structure and accessibility of the transcriptional machinery to DNA, thereby regulating gene expression [127]. Histone acetylation typically correlates with active gene transcription, as it loosens the chromatin structure and allows transcription factors to access DNA. In contrast, histone deacetylation is associated with gene repression [123]. In PDEECs, aberrant histone deacetylation mediated by histone deacetylases (HDACs) can result in the silencing of tumor suppressor genes, leading to cancer progression. Inhibition of HDACs has been explored as a therapeutic strategy for endometrial cancers [128]. HDAC inhibitors (HDACi) are being studied for their ability to restore the expression of silenced tumor suppressor genes and induce cell cycle arrest, apoptosis, and differentiation in cancer cells [129]. Histone methylation can either activate or repress gene transcription, depending on the specific modified residues and type of methylation (mono-, di-, or tri-methylation) [130]. For example, trimethylation of H3K27 (histone 3 at lysine 27) is a repressive mark often catalyzed by EZH2, a component of the Polycomb Repressive Complex 2 (PRC2) [123]. Overexpression of EZH2 and excessive H3K27 trimethylation have been linked to aggressive tumor behavior in PDEECs by silencing tumor suppressor genes and maintaining cells in an undifferentiated state [124].

Noncoding RNAs (ncRNAs), including microRNAs (miRNAs) and long noncoding RNAs (lncRNAs), regulate gene expression at the post-transcriptional level and play a significant role in endometrial cancer pathogenesis [124]. ncRNAs can act as either oncogenes or tumor suppressors. miRNAs are small RNA molecules that regulate gene expression by binding to the 3’-untranslated regions (UTRs) of target messenger RNAs (mRNAs), leading to their degradation or inhibition of translation [131]. Dysregulation of miRNAs is frequently observed in PDEECs. For example, the miR-200 family regulates epithelial–mesenchymal transition (EMT), a process critical for cancer metastasis [123]. In PDEECs, downregulation of miR-200 has been associated with increased EMT, promoting invasiveness and metastatic potential. miR-34, often regarded as a tumor suppressor miRNA, is frequently downregulated in PDEECs, leading to the activation of oncogenic pathways and contributing to tumor progression [123].

Long non-coding RNAs (lncRNAs) are large RNA molecules that play diverse roles in gene regulation, including chromatin remodeling, transcriptional control, and post-transcriptional processing [53]. In PDEECs, dysregulated lncRNAs have been implicated in tumor progression, resistance to apoptosis, and immune evasion. Overexpression of HOTAIR, a well-known oncogenic lncRNA, has been linked to aggressive features of various cancers, including poor differentiation, in various cancers [132]. In endometrial cancer, HOTAIR modulates chromatin remodeling and gene silencing, thereby promoting tumor growth and metastasis. Given the importance of epigenetic alterations in PDEECs, targeting epigenetic modifications has emerged as a promising therapeutic strategy [123]. Drugs that reverse abnormal DNA methylation and histone modifications, such as DNA methyltransferase inhibitors (DNMTi) and HDACi, are being explored for the treatment of endometrial cancers [133]. DNMT inhibitors (e.g., azacitidine and decitabine) work by demethylating the promoter regions of tumor suppressor genes, restoring their expression and inducing tumor cell death [134]. HDAC inhibitors (e.g., vorinostat and romidepsin) modulate histone acetylation, leading to chromatin relaxation and transcriptional activation of genes involved in cell cycle arrest and apoptosis. They are being tested in clinical trials for their ability to reactivate silenced tumor suppressor genes and inhibit cancer cell growth in poorly differentiated tumors [135]. Clinical trials investigating these agents in endometrial cancer have reported variable outcomes. For instance, a study evaluating the HDAC inhibitor vorinostat demonstrated limited efficacy as monotherapy in recurrent endometrial cancer [1,132]. However, combining epigenetic drugs with other treatments, such as hormonal therapy or immune checkpoint inhibitors, is being explored to enhance their therapeutic efficacy [134]. Ongoing research aims to identify biomarkers that predict the response to epigenetic therapies and to optimize combination strategies, potentially improving outcomes for patients with PDEECs. According to ref. [123], epigenetic modifications of PDEECs have profound clinical implications. Methylation patterns and ncRNA expression profiles can serve as biomarkers for PDEEC prognosis. For example, hypermethylation of MLH1 and PTEN or overexpression of HOTAIR can be associated with worse outcomes [136]. Epigenetic therapies that aim to reverse abnormal methylation or histone modifications hold promise as targeted treatments for patients with PDEECs, particularly for those resistant to conventional therapies [124]. Epigenetic drugs may also be used in combination with immune checkpoint inhibitors or chemotherapy to enhance treatment efficacy. By modulating the epigenome, these drugs can sensitize cancer cells to immune responses and traditional chemotherapeutic agents [133]. Thus, epigenetic modifications play a critical role in the progression and aggressiveness of PDEECs. Aberrant DNA methylation, histone modifications, and non-coding RNA dysregulation contribute to the silencing of tumor suppressor genes, activation of oncogenic pathways, and increased genomic instability [137]. These changes not only serve as biomarkers for prognosis but also offer potential targets for epigenetic therapies. With further research, epigenetic modifications may provide a path toward personalized treatment strategies that improve outcomes in patients with high-grade, aggressive endometrial cancers.

## 5. Molecular Pathways Implicated in PDEECs

### 5.1. PI3K/AKT/mTOR Pathway

The PI3K/AKT/mTOR pathway is a critical signaling cascade involved in cell proliferation, survival, metabolism, and growth [138]. Dysregulation of this pathway is often observed in PDEECs and plays a substantial role in tumor aggressiveness, progression, and resistance to therapy [139]. The PI3K/AKT/mTOR pathway is frequently activated by mutations in genes encoding components of the pathway, including PIK3CA, PTEN, and AKT, leading to uncontrolled cellular proliferation and survival, thus contributing to the aggressive nature of poorly differentiated tumors [138]. As shown in Figure 5, the PI3K/AKT/mTOR pathway is typically activated by growth factors binding to receptor tyrosine kinases (RTKs), such as EGFR or IGFR, which then activate phosphoinositide 3-kinase (PI3K). PI3K converts PIP2 to PIP3, which activates AKT, a central kinase in this pathway [140]. AKT activation leads to downstream activation of mTOR (mechanistic target of rapamycin), a kinase that promotes cellular growth and metabolism through the regulation of protein synthesis, cell cycle progression, and inhibition of apoptosis [139]. In a normal cellular context, this pathway is tightly regulated, particularly by phosphatase and tensin homolog (PTEN), a tumor suppressor that dephosphorylates PIP3 back to PIP2, thereby counteracting PI3K activity [58].

In PDEECs, various genetic alterations lead to the constitutive activation of the PI3K/AKT/mTOR pathway [139]. PTEN acts as a brake on the PI3K/AKT pathway, and its loss leads to increased PIP3 levels, resulting in continuous activation of AKT and subsequent mTOR signaling [139]. This uncontrolled signaling contributes to resistance to apoptosis, a hallmark of aggressive cancer phenotypes [97]. Mutations in AKT1 have also been identified in PDEECs, leading to AKT activation and downstream signaling independent of upstream inputs. This alteration further compounds the effects of PIK3CA and PTEN mutations, making the pathway hyperactive [141].

**Figure 5 cells-14-00382-f005:**
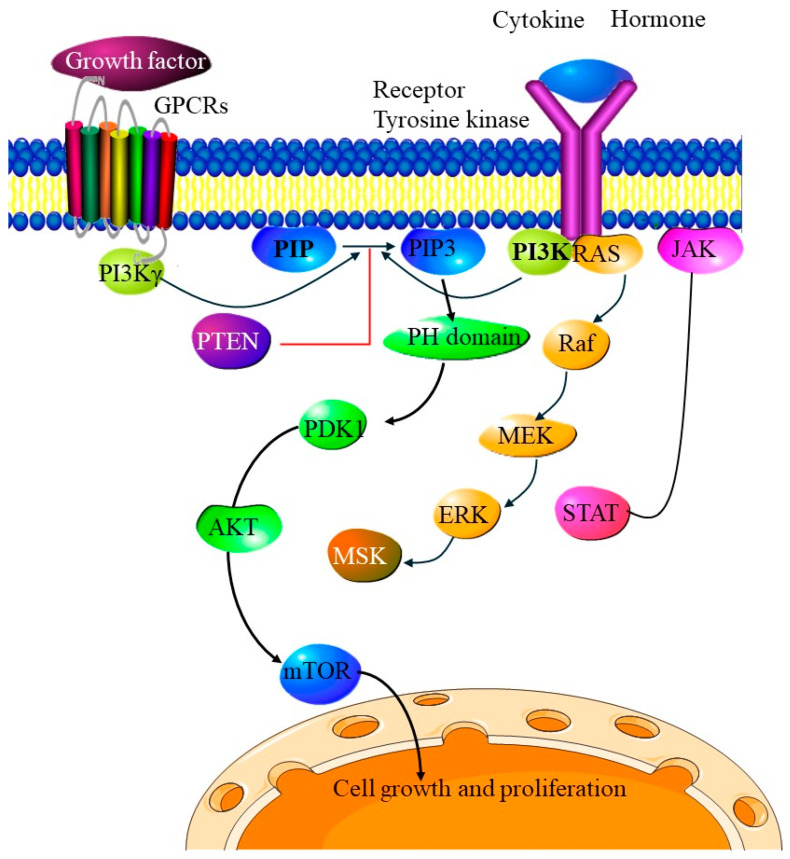
Overview of the PI3K/AKT/mTOR pathway. In healthy cells, this pathway is tightly regulated, with phosphatase and tensin homolog (PTEN) acting as a critical suppressor by dephosphorylating PIP3 back to PIP2, thereby reducing PI3K/AKT activity. Mutations or dysregulation of genes such as PIK3CA, PTEN, and AKT are common in PDEECs, where the PI3K/AKT/mTOR pathway contributes to uncontrolled cell growth, resistance to apoptosis, and enhanced survival. Consequently, this pathway is a significant focus for PDEEC research, with inhibitors targeting PI3K, AKT, and mTOR showing potential in treating various tumors, especially those that exhibit high pathway activation owing to genetic alterations (figure adapted from [142]).

The PI3K/AKT/mTOR pathway plays several roles in the progression of PDEECs. Activation of this pathway promotes cell cycle progression by upregulating cyclin D1 and downregulating p27, a cyclin-dependent kinase inhibitor [143]. This allows cancer cells to proliferate unchecked, contributing to the rapid growth of poorly differentiated cancers [144]. AKT activation leads to the phosphorylation and inactivation of pro-apoptotic factors, such as BAD and caspase-9. This pathway also inhibits the activity of FOXO transcription factors, which would otherwise promote apoptosis and cell cycle arrest [139]. Consequently, PDEECs are less likely to undergo programmed cell death, contributing to resistance to standard therapies that rely on apoptotic mechanisms [145]. The mTOR complex (mTORC1) promotes metabolic changes that enhance the energy supply and biomass required for rapid cell division, including glycolysis, lipid synthesis, and protein synthesis [146]. By reprogramming cellular metabolism, the PI3K/AKT/mTOR pathway supports the growth and survival of highly proliferative, poorly differentiated tumor cells [147].

Dysregulation of the PI3K/AKT/mTOR pathway in PDEECs makes it an attractive therapeutic target. Various inhibitors targeting different components of this pathway have been developed and are currently under investigation in clinical trials for endometrial cancer treatment [148]. These drugs aim to reduce PI3K activity, thereby decreasing downstream AKT and mTOR signaling; however, they may have significant side effects owing to PI3K’s widespread role in normal cellular function [149]. AKT inhibitors, which block the pathway further downstream, may also offer a more focused approach to reducing survival and proliferation signals in cancer cells. However, resistance mechanisms, such as compensatory upregulation of other survival pathways, remain challenging [150]. Rapalogs, such as everolimus and temsirolimus, inhibit mTORC1 and have shown some efficacy in treating endometrial cancer. Dual mTORC1/mTORC2 inhibitors are also under study, as they may provide more comprehensive suppression of mTOR signaling by preventing the feedback activation of AKT [151]. Combination therapies involving PI3K/AKT/mTOR inhibitors with chemotherapy or immune checkpoint inhibitors are being explored to counteract pathway redundancy and resistance mechanisms [139]. This approach is expected to improve outcomes by targeting both cancer cells and the immune environment, which often suppresses anti-tumor responses in PDEECs [8].

Despite the therapeutic potential of targeting the PI3K/AKT/mTOR pathway, several challenges remain [123]. Tumor cells often develop resistance to inhibitors of the PI3K/AKT/mTOR pathway through compensatory activation of alternative pathways, such as MAPK/ERK and Wnt/β-catenin [152]. This adaptive resistance diminishes the efficacy of monotherapy. Inhibition of mTORC1 can lead to feedback activation of AKT, which can paradoxically enhance tumor cell survival. This necessitates the development of dual inhibitors that can effectively suppress both mTOR complexes and AKT activation [153]. Given the role of the PI3K/AKT/mTOR pathway in normal cellular processes, inhibitors may have significant side effects, limiting their dosage and therapeutic window. Thus, the PI3K/AKT/mTOR pathway plays a central role in the pathogenesis and progression of PDEECs, making it a critical target for therapeutic interventions [123]. However, the inherent complexity and redundancy of this pathway, along with its importance in normal cellular function, present challenges for effective targeting [139]. Advances in combination therapy strategies as well as the development of inhibitors with improved specificity may offer promising avenues for improving the outcomes of patients with PDEECs [150]. Future research should continue to focus on overcoming resistance mechanisms and minimizing side effects to harness the therapeutic potential of PI3K/AKT/mTOR pathway inhibition while preserving normal cellular processes.

#### Immunosuppressive Effects of PI3K/AKT/mTOR Inhibitors

The immunosuppressive effects of PI3K/AKT/mTOR inhibitors are significant because of their critical role in immune cell function. Inhibition of this pathway can lead to reduced T-cell activation and proliferation, impaired B-cell development, and decreased natural killer cell activity, ultimately compromising the immune response and increasing susceptibility to infections [154]. Clinical trials have reported elevated rates of infections, including pneumonia and sepsis, in patients receiving these inhibitors, highlighting the need for vigilant monitoring and potential prophylactic measures [154]. Furthermore, the balance between the antitumor effects of these inhibitors and their immunosuppressive consequences is particularly challenging in patients with pre-existing immune deficiencies [155]. Future research should focus on strategies to mitigate these adverse effects while preserving therapeutic efficacy against tumors [155].

### 5.2. p53 Pathway Dysfunction

The p53 pathway plays a central role in maintaining genomic integrity and mediating cell cycle regulation, DNA repair, and apoptosis in response to cellular stress [156]. In PDEECs, dysfunction of this pathway, often due to mutations in TP53, is a prominent feature with significant implications for tumor progression, treatment response, and patient outcomes [150]. When activated, p53 binds to specific DNA sequences and regulates genes involved in cell cycle arrest (e.g., p21), DNA repair (e.g., GADD45), and apoptosis (e.g., BAX and PUMA) [139]. This regulation is essential for preventing genomic instability and controlling unregulated cell proliferation.

As shown in Figure 6, p53 dysfunction impairs the DNA damage response, allowing cells with significant DNA damage to evade apoptosis. This contributes to genomic instability, one of the hallmarks of cancer, and facilitates further mutations, fueling tumor progression in PDEECs [37]. According to ref. [123], studies show that p53 inactivation correlates with the high-grade aggressive nature of these cancers, distinguishing them from more differentiated, lower-grade endometrial cancers [157]. TP53 mutations are negative prognostic factors in endometrial cancer, particularly in PDEECs. The presence of p53 dysfunction suggests a more aggressive disease course and may predict reduced survival [157]. Immunohistochemical testing for p53 protein overexpression or loss, often a surrogate for TP53 mutations, is used to assess cancer prognosis and guide therapeutic decisions. The dysfunction of p53 is also linked to resistance to conventional therapies [150]. Chemotherapies that rely on functional p53-induced apoptosis (such as DNA-damaging agents) tend to be less effective in p53-mutated tumors because these cells cannot undergo apoptosis as readily. This resistance poses a significant challenge in managing PDEECs, as standard therapies may not achieve optimal results [158]. Research on immunotherapies specifically for TP53-mutant endometrial cancers is ongoing. p53-mutant tumors often exhibit higher mutational burdens, which may be more immunogenic, making them suitable candidates for immune checkpoint inhibitors [37].

Ongoing research aims to better understand the downstream effects of p53 dysfunction in PDEECs. Advanced genomic studies, such as The Cancer Genome Atlas (TCGA), continue to elucidate the genetic landscape of these tumors, providing insights into the complex interplay between TP53 mutations and other pathways, such as PI3K/AKT/mTOR, which are frequently co-altered [150]. Studies have also focused on understanding how the tumor microenvironment in PDEECs interacts with p53-dysfunctional cells, with the goal of identifying potential targets for combination therapies.

### 5.3. Mismatch Repair (MMR) Deficiency and Microsatellite Instability (MSI)

Mismatch Repair (MMR) Deficiency and Microsatellite Instability (MSI) play significant roles in the progression, molecular landscape, and clinical management of PDEECs [160]. MMR deficiency, often resulting in MSI, contributes to the accumulation of mutations, particularly in repetitive DNA sequences, which increases genomic instability—a hallmark of many cancers, including high-grade endometrial cancers [161]. Understanding MMR and MSI is critical for defining the prognosis, therapeutic approaches, and potential for personalized treatments for patients with PDEECs [95]. As shown in Figure 7, the MMR system is a DNA repair pathway responsible for correcting DNA replication errors, specifically base–base mismatches and insertion–deletion loops, that occur during DNA replication [162]. Core MMR proteins, such as MLH1, MSH2, MSH6, and PMS2, work in concert to identify and repair replication errors. Loss or mutation of any of these proteins impairs the MMR system, leading to an accumulation of mutations across the genome, particularly in regions of repetitive DNA known as microsatellites [95]. MMR deficiency often arises from either somatic mutations or, in some cases, hypermethylation, such as MLH1 promoter methylation, which silences gene expression [162]. MSI is a molecular consequence of MMR deficiency, characterized by length variations in microsatellite regions due to uncorrected replication errors [161]. MSI is divided into categories based on the number of altered microsatellite markers: MSI-high (MSI-H) indicates a significant degree of instability, whereas the MSI-low and microsatellite stable (MSS) categories show less or no instability, respectively [160]. In PDEECs, MSI-H status is often indicative of a distinct molecular profile associated with a high mutational burden and specific mutational patterns that drive cancer progression [95].

PDEECs with MMR deficiency exhibit increased genomic instability, leading to the accumulation of mutations across various cancer-related genes, including PTEN, PIK3CA, and ARID1A [160]. This instability can fuel rapid tumor progression and contribute to the aggressive phenotype observed in high-grade endometrial cancers. The high mutational burden and genomic instability in MSI-H PDEECs promote significant tumor heterogeneity [42]. This variability within the tumor population can contribute to therapy resistance, as diverse subpopulations of cancer cells may respond differently to treatment [161]. In endometrial cancers, MSI-H status is often associated with a better overall prognosis than MSI-low or MSS status, possibly due to the increased immunogenicity of MSI-H tumors, which elicits a more robust immune response [160]. However, the exact prognostic value of PDEECs requires further research, as the aggressive nature of poorly differentiated tumors may diminish this survival advantage.

MMR-deficient and MSI-H tumors exhibit high TMB, generating neoantigens that make them more visible to the immune system [160]. This property makes MSI-H PDEECs particularly responsive to immune checkpoint inhibitors (ICIs), such as pembrolizumab, which blocks PD-1/PD-L1 pathways and enhances the immune response against the tumor [160]. The FDA has approved ICIs for MSI-H or MMR-deficient solid tumors, regardless of their tissue origin, making this an important therapeutic option for MSI-H PDEECs [164]. The use of pembrolizumab has shown promising results in several studies, providing durable responses in MMR-deficient endometrial cancers [161]. Researchers are exploring combination therapies involving ICIs and other agents to further enhance the immune response in MSI-H tumors [165]. For example, combining ICIs with DNA-damaging agents or targeted therapies (such as PI3K or AKT inhibitors) could enhance tumor susceptibility to immune attack [166].

Testing for MMR deficiency and MSI status is now standard in endometrial cancer, especially for high-grade and poorly differentiated tumors, owing to the therapeutic implications of immunotherapy [167]. Diagnostic methods include immunohistochemistry (IHC) to assess MMR protein expression, and PCR-based assays or next-generation sequencing (NGS) to determine MSI status [161]. Determining MMR and MSI status in PDEECs allows for the personalization of therapy. For example, MSI-H or MMR-deficient tumors are likely candidates for ICI therapy, whereas MSS or MMR-proficient tumors may benefit more from alternative therapeutic approaches, such as hormone therapy or targeted molecular inhibitors [160]. Although ICIs show promise for MSI-H PDEECs, resistance mechanisms can develop, limiting the effectiveness of treatment over time. Research is underway to identify biomarkers that predict the response to immunotherapy as well as alternative pathways that could be targeted to overcome resistance [168]. Ongoing clinical trials are exploring the use of ICIs in combination with other treatments, such as chemotherapy, targeted therapy, and radiation, to improve their efficacy in MSI-H endometrial cancers. These studies aimed to determine the optimal combination regimens and sequencing of therapies for better clinical outcomes [161].

Further research into biomarkers associated with MSI-H and MMR deficiency in PDEECs could enhance diagnostic precision and guide the selection of targeted therapies [161]. Additionally, understanding the interplay between MMR deficiency and other pathways (such as PI3K/AKT) may reveal new therapeutic targets [138]. Thus, MMR deficiency and MSI play critical roles in the development and management of PDEECs [160]. MSI-H and MMR-deficient tumors are characterized by genomic instability, high mutational burden, and a robust immune response, making them suitable candidates for immunotherapy [167]. Diagnostic testing for MSI and MMR status is essential for personalized treatment planning, with immunotherapy showing particular promise for MSI-H PDEECs [160]. As research continues, advancements in biomarker discovery and combination therapies may further improve clinical outcomes for patients with aggressive tumors [123].

## 6. Biomarkers and Molecular Targets in PDEECs

The genomic landscape of PDEECs has allowed for the identification of molecular markers essential for understanding the biology of the disease and for guiding diagnosis, prognosis, and therapy [169]. These markers are involved in key signaling pathways related to tumor initiation, progression, and resistance to therapy. They provide crucial insights into tumor behavior, prognosis, and therapeutic options [170]. Mutations in PDEECs are driven by a combination of genetic mutations and epigenetic modifications (Table 4 [169]). Genetic mutations in PDEECs involve alterations in tumor suppressor genes, oncogenes, and DNA repair pathways, leading to uncontrolled cell proliferation and genomic instability. Examples of the most common genetic mutations in PDEECs are TP53, PPP2R1A, FBXW7, PIK3CA, PTEN, CTNNB1, HER2, and KRAS [171]. Unlike genetic mutations, epigenetic changes do not alter the DNA sequence but regulate gene expression through mechanisms such as DNA methylation, histone modifications, and non-coding RNAs [170,171]. Epigenetic modifications play a significant role in the initiation and progression of PDEECs by silencing tumor suppressor genes and activating oncogenic pathways. The most common epigenetic modification in PDEECs is the ARID1A gene [169]. Understanding the interplay between genetic and epigenetic alterations is crucial for deciphering tumor behavior, prognosis, and therapeutic vulnerabilities [169,170].

TP53 mutations are highly prevalent in high-grade PDEECs, especially in serous and other aggressive subtypes [171]. TP53 mutations are found in up to 90% of serous endometrial carcinomas and in approximately 20–30% of high-grade endometrioid carcinomas [160]. Given its association with chemoresistance, TP53 mutations can guide therapeutic decision-making and identify patients who may require aggressive treatment [158].

PPP2R1A mutations are primarily identified in aggressive subtypes of endometrial cancer, including serous and poorly differentiated endometrioid carcinomas [160]. These mutations are frequently found in the exon 5 region of PPP2R1A, which encodes part of the scaffold subunit of PP2A phosphatase, a critical regulator of cell signaling, growth, and apoptosis [172]. Alterations in PPP2R1A, often involving missense mutations, impair the assembly or stability of the PP2A complex, compromising its tumor-suppressive functions [172]. In high-grade EECs, PPP2R1A mutations are observed in approximately 20–40% of cases, often co-occurring with other mutations, such as TP53, which may enhance tumor aggressiveness [123]. Their relatively high prevalence in poorly differentiated tumors suggests that PPP2R1A mutations contribute to the progression and poor prognosis observed in these subtypes [169]. PPP2R1A mutations are often found alongside TP53 mutations, further promoting an aggressive tumor phenotype. Although not as frequently mutated as TP53, PPP2R1A mutations indicate a high-risk profile [123]. As prognostic biomarkers, PPP2R1A mutations offer insights into disease severity and survival prospects, whereas, as predictive biomarkers, they provide potential targets for developing innovative therapeutic approaches in managing PDEECs [160]. Targeted therapies that modulate PP2A-related pathways remain elusive, and no specific treatment targeting PPP2R1A mutations has yet been established [173].

FBXW7 mutations occur in approximately 5–20% of high-grade endometrial tumors, with a higher prevalence in tumors displaying aggressive histologic features. FBXW7 is a tumor suppressor that targets several oncoproteins, including MYC and cyclin E, for degradation [174]. Loss of FBXW7 function results in the accumulation of oncoproteins that drive uncontrolled proliferation and resistance to apoptosis. Mutations typically occur in the WD40 domain, a region critical for the protein’s ability to recognize and bind to its substrates, which prevents the degradation of oncogenic targets [160]. FBXW7 mutations are often found alongside mutations in other tumor suppressor genes, such as TP53, suggesting that FBXW7 loss may play a collaborative role in driving the aggressive nature of these cancers [175]. Given their association with high-grade tumors, FBXW7 mutations have attracted attention as biomarkers for aggressive diseases and potential prognostic indicators in PDEECs [176]. FBXW7 mutations are associated with poor prognosis and may indicate chemoresistance, suggesting a potential benefit from MYC or NOTCH pathway inhibitors. FBXW7 status may help in risk stratification and guide the exploration of targeted treatments [174].

PIK3CA mutations are found in approximately 20–40% of endometrial cancers, predominantly in high-grade tumors and in cancers classified as serous or clear cell subtypes, which are more aggressive and poorly differentiated [67]. PIK3CA mutations lead to pathway hyperactivation, which supports cell proliferation, metabolic changes, and survival. These mutations frequently co-occur with PTEN loss, creating a synergistic effect that drives tumor progression [177]. PIK3CA mutations are prognostic and predictive markers. Studies have shown that these mutations correlate with shorter overall survival and progression-free survival in patients with endometrial cancer, especially those with high-grade tumors [69].

Mutations or deletions in PTEN are common in endometrial cancers and often co-occur with PIK3CA mutations, occurring in up to 80% of endometrioid endometrial carcinomas and frequently associated with poorly differentiated, aggressive tumors [178]. In PDEECs, PTEN mutations have significant prognostic and therapeutic implications, underscoring their potential utility as biomarkers [179]. PTEN mutations are generally associated with poorer prognosis, particularly in high-grade, poorly differentiated tumors [178]. Loss of PTEN function is correlated with increased tumor aggressiveness, higher recurrence rates, and decreased survival. This poor prognosis aligns with increased PI3K pathway activity, which supports malignant behaviors, such as invasion and metastasis [123]. PTEN-deficient tumors often exhibit resistance to standard chemotherapies owing to their enhanced survival signaling pathways. The loss of PTEN activates downstream pathways that increase drug efflux, reduce apoptosis, and support cell survival, making these tumors less responsive to conventional treatments, such as platinum-based chemotherapies [180]. PTEN mutations indicate potential responsiveness to PI3K/AKT/mTOR pathway inhibitors. Thus, PTEN mutations serve as significant biomarkers in PDEECs, representing both prognostic and therapeutic values. By identifying PTEN loss or mutation, clinicians can better predict disease progression, treatment response, and survival outcomes [181].

**Table 4 cells-14-00382-t004:** The most common genetic and epigenetic alterations in PDEECs.

Genetic Alterations	Carcinoma (10–20%)	Clear Cell Carcinoma (<5%)	Carcinosarcoma (<5%)	Genetic Mutation	Epigenetic Mutation	References
*TP53*	57.7–92%	29–46%	64.3–91%	✅	❌	[182]
*PPP2R1A*	15.4–43.2%	15.9–36%	0–28.1%	✅	❌	[183]
*FBXW7*	17.3–29%	7.9–25%	39%	✅	❌	[184]
*PTEN*	2.7–22.5%	11–21%	19–33.3%	✅	❌	[183]
*ARID1A*	0–10.8%	15–21%	12–23.8%	✅	✅	[184]
*PIK3CA*	10–47%	23.8–36%	17–35%	✅	❌	[183]
*CTNNB1*	2.7%	0%	4.8%	✅	❌	[185]
*KRAS*	2–8%	12–16.7%	14%	✅	❌	[173]
*HER2*	17–44%	12–50%	0–20%	✅	❌	[186]

Percentages in the header refer to all EC cases; percentages in the table refer to each PDEEC case.

ARID1A mutations in EECs are linked to high-grade, poorly differentiated tumors, which are more likely to exhibit aggressive clinical behaviors, including rapid growth and metastasis [8]. Thus, ARID1A mutations can serve as prognostic markers for identifying patients with potentially poor outcomes who may require more intensive therapeutic approaches [187]. Since ARID1A-mutant tumors often show increased microsatellite instability (MSI) and other forms of genomic instability, they can be used as biomarkers to predict disease behavior and progression. Tumors with ARID1A loss are more likely to have DNA repair deficiencies, which also correlate with tumor aggressiveness and higher recurrence rates [188].

Due to compromised DNA repair mechanisms, ARID1A-mutant tumors show increased sensitivity to DNA-damaging agents, such as platinum-based chemotherapy [84]. This sensitivity highlights ARID1A’s predictive value in identifying patients who might benefit more from treatments that exploit DNA repair deficiencies [80]. Given the association between ARID1A loss and MSI, ARID1A-mutant cancers are more likely to exhibit high neoantigen loads, making them potential candidates for immune checkpoint inhibitors [169]. MSI-high cancers have shown promising responses to immunotherapies, as the increased mutation load in these tumors can provoke an immune response that checkpoint inhibitors can amplify [150]. The impairment of homologous recombination due to ARID1A loss suggests that PARP inhibitors, which target single-strand DNA break repair, may be effective against ARID1A-mutant cancers [8]. In the absence of functional ARID1A, cells rely more heavily on alternative DNA repair pathways, making them particularly vulnerable to PARP inhibition [150]. Studies exploring ARID1A-deficient cells in combination with PARP inhibitors have shown potential synthetic lethality, which is a promising therapeutic approach [84]. ARID1A-mutant cancers, particularly those with upregulated PI3K/AKT signaling, may respond to targeted therapies that inhibit this pathway [150]. PI3K inhibitors can counteract the enhanced survival signals conferred by ARID1A mutations, offering a targeted approach that could improve therapeutic outcomes [189]. Thus, ARID1A mutations represent a promising biomarker for prognosis as well as a predictor of therapeutic response. As research progresses, ARID1A holds promise not only as a biomarker but also as a potential target for novel therapeutic strategies for PDEECs [84].

Mutations in CTNNB1 are frequently observed in low- and intermediate-grade endometrioid carcinomas, but are less common in PDEECs [169]. CTNNB1 encodes β-catenin, a key component of the WNT signaling pathway. When CTNNB1 is mutated, β-catenin accumulates in the nucleus and activates the transcription of genes involved in cell proliferation and survival [95]. CTNNB1 mutations are typically associated with lower-grade tumors; however, when present in PDEECs, they may indicate responsiveness to therapies targeting components of the WNT pathway [95]. CTNNB1 mutations are significant biomarkers in PDEECs, with implications for prognosis, treatment stratification, and development of targeted therapies [150]. By driving the Wnt/β-catenin pathway, these mutations contribute to tumor aggressiveness, invasiveness, and heterogeneity, making them markers of poor prognosis and potential therapeutic targets [89]. As research advances, the integration of CTNNB1 mutation status into clinical practice holds promise for improving personalized treatment strategies for endometrial cancers, particularly when used alongside multi-targeted approaches and comprehensive biomarker panels [91]. Continued clinical trials and advancements in Wnt pathway inhibition are essential to fully exploit the potential of CTNNB1 as a biomarker and therapeutic target in PDEECs [190]. CTNNB1 mutations are associated with advanced disease stages, high tumor grade, and poor prognosis [95]. This correlation suggests that CTNNB1 can serve as a prognostic biomarker, helping clinicians identify patients with high-risk tumors who may require more intensive monitoring and aggressive treatment approaches [169].

Considering that CTNNB1 mutations promote EMT and cellular invasiveness, tumors harboring these mutations are often more aggressive and have a higher likelihood of metastasis [95]. Identifying CTNNB1 mutations in EECs can aid in stratifying patients based on their metastatic risk and in guiding decisions regarding surgical margins, lymph node assessment, and adjuvant therapy [169]. The Wnt/β-catenin pathway, which is activated by CTNNB1 mutations, presents an opportunity for targeted therapies. Small-molecule inhibitors and antibodies targeting various components of the Wnt pathway, including β-catenin, are under investigation in preclinical and early-phase clinical trials [190]. In EECs with CTNNB1 mutations, such therapies may inhibit tumor growth and invasiveness by blocking downstream oncogenic signals [95]. Evidence suggests that Wnt pathway activation can influence immune cell infiltration within tumors, potentially impacting the tumor immune microenvironment [191]. In the context of immunotherapy, CTNNB1 mutations may help predict which patients might benefit from combination therapies that include immune checkpoint inhibitors alongside Wnt pathway modulators, enhancing antitumor immune responses [160]. Given the frequent co-occurrence of CTNNB1 and PIK3CA mutations, dual inhibition of the Wnt and PI3K/AKT pathways may be effective in PDEECs [95]. Early studies have shown that co-targeting these pathways can reduce tumor growth and induce apoptosis more effectively than targeting either pathway alone [124].

HER2 (human epidermal growth factor receptor 2) is a proto-oncogene encoding a transmembrane receptor tyrosine kinase involved in cell proliferation, differentiation, and survival [124]. HER2 dysregulation through gene amplification, overexpression, or specific mutations has been shown to drive tumorigenesis and progression in several cancers [25]. In PDEECs, HER2 mutations and amplifications are commonly associated with serous endometrial carcinomas, a high-grade aggressive subtype [192]. Overexpression or mutation of HER2 results in constitutive activation of downstream signaling pathways, leading to aggressive tumor growth and metastasis [193]. HER2 overexpression is a promising biomarker for targeted therapies, particularly HER2-targeted antibodies, such as trastuzumab. HER2 status can be used to identify patients who may benefit from anti-HER2 therapies, as shown in clinical trials [25]. HER2 overexpression and amplification in PDEECs are correlated with advanced disease stage, higher tumor grade, and decreased overall survival [194]. In studies comparing HER2-positive and HER2-negative endometrial tumors, HER2 positivity was associated with shorter progression-free survival, highlighting its utility as a prognostic marker for high-risk aggressive endometrial cancers [195]. This association with poor prognosis suggests that HER2 testing in EECs could aid in stratifying patients based on risk, allowing for closer monitoring and potentially more aggressive management of patients with HER2-positive disease [150].

HER2 alterations not only serve as prognostic markers but also indicate potential responsiveness to HER2-targeted therapies, similar to their role in breast cancer [196]. Anti-HER2 agents, including trastuzumab (an anti-HER2 monoclonal antibody) and HER2-targeted tyrosine kinase inhibitors (TKIs), have demonstrated efficacy in patients with HER2-positive endometrial cancer patients [25]. Trastuzumab has shown promising results in clinical trials when combined with chemotherapy for HER2-positive endometrial serous carcinomas, leading to improved progression-free survival compared to chemotherapy alone [150]. Thus, the predictive value of HER2 as a biomarker is significant in PDEECs, where HER2-targeted therapies may improve clinical outcomes in an otherwise difficult-to-treat population [192]. Thus, HER2 mutations and amplifications are significant biomarkers of PDEECs and provide valuable prognostic and predictive information. Their presence is associated with aggressive tumor behavior, poor prognosis, and potential responsiveness to HER2-targeted therapies [124].

In PDEECs, KRAS mutations have been identified at similar rates, with an increased prevalence noted in aggressive subtypes, such as serous and high-grade endometrioid cancers, both of which often exhibit poor differentiation [197]. KRAS mutations serve as critical biomarkers in PDEECs, correlating with aggressive disease, poor prognosis, and resistance to conventional therapies [160]. Their presence not only guides the identification of high-risk patients but also informs targeted therapeutic strategies that may improve outcomes in this challenging subset of endometrial cancers [4]. Although the direct targeting of KRAS remains complex, ongoing research and combination therapies offer promising approaches for integrating KRAS as a valuable marker in personalized oncology [169]. Targeting downstream signaling pathways, such as MEK inhibitors, may offer therapeutic benefits for KRAS-mutant tumors. MEK inhibitors, including trametinib and selumetinib, have shown efficacy in other KRAS-mutant cancers and are currently being investigated for their potential use in endometrial cancers [198].

KRAS mutations in PDEECs are generally associated with poor clinical outcomes, including higher recurrence rates and reduced overall survival [199]. Patients with KRAS-mutant tumors often exhibit resistance to standard treatments, such as hormonal therapies and chemotherapies, owing to the aggressive phenotype driven by these mutations [124]. KRAS mutations, particularly in conjunction with other oncogenic mutations, also correlate with poor responses to radiotherapy, limiting treatment options and complicating management strategies for patients with advanced disease [200]. Screening for KRAS mutations in endometrial cancers may assist in identifying patients at high risk of aggressive disease progression [201]. The integration of KRAS mutation status into clinical practice could improve risk stratification, allowing for more individualized treatment planning and closer monitoring of patients with high-risk profiles [199]. Additionally, the identification of KRAS mutations at earlier stages of the disease could help guide the choice of initial treatment strategies, potentially improving survival outcomes by adapting interventions to the tumor’s molecular profile from the onset [150].

The discovery of molecular alterations in PDEECs plays a unique and critical role in the molecular landscape of PDEECs, shaping their prognosis, therapeutic response, and potential for targeted treatment strategies [202]. By identifying specific biomarkers and molecular targets, treatments can be tailored to individual tumors, improving both prognosis and patient outcomes. TP53, PPP2R1A, and FBXW7 mutations are associated with poor prognosis and can inform clinicians about the likely clinical outcomes and risk of recurrence [150]. PIK3CA, PTEN, HER2, and KRAS mutations suggest opportunities for targeted therapy. HER2 amplification is actionable through HER2-targeted drugs, whereas PI3K pathway alterations may benefit from pathway-specific inhibitors [124]. Research on ARID1A and CTNNB1 mutations is ongoing, as they offer potential predictive value for immunotherapy and targeted therapies. Future studies and clinical trials are essential to translate these genetic insights into effective and personalized treatment plans for patients with PDEECs [203]. Integrating these biomarkers into clinical practice will enable better-informed decision-making and help advance personalized treatment approaches for endometrial cancer.

## 7. Therapeutic Targets

### 7.1. Targeting the PI3K/AKT/mTOR Pathway

Targeting the PI3K/AKT/mTOR pathway in PDEECs offers a promising therapeutic strategy, given the frequent alterations observed within this pathway in these aggressive tumor subtypes. PI3K inhibitors target specific isoforms or the catalytic subunit of PI3K [66]. Alpelisib, a PI3Kα-specific inhibitor, has shown efficacy in cancers harboring PIK3CA mutations, including endometrial cancers, by reducing AKT phosphorylation and downstream signaling [204]. Selective targeting of the PI3Kα isoform is beneficial for reducing the toxicities associated with pan-PI3K inhibition. In endometrial cancer, BYL719 (alpelisib) has been studied in clinical trials, revealing moderate efficacy in cases with specific PIK3CA mutations [205]. However, single-agent PI3K inhibition often results in limited response durations owing to pathway feedback mechanisms and compensatory activation of parallel signaling pathways [206]. AKT inhibitors, such as Ipatasertib and Capivasertib, have also been explored in preclinical and clinical studies involving endometrial cancers. AKT inhibition may be particularly effective in tumors with both PTEN loss and PIK3CA mutations, as these cancers often demonstrate AKT pathway hyperactivation [124]. Targeting AKT reduces the phosphorylation of mTOR and other downstream targets, curbing cell growth and inducing apoptosis. Despite the promise shown in preclinical studies, clinical results in endometrial cancer have shown mixed efficacy, with combination therapy often necessary to enhance responses [150].

mTOR inhibitors, such as Everolimus and Temsirolimus, directly target the mTORC1 complex and have demonstrated antitumor effects in endometrial cancers by inhibiting cell growth and protein synthesis [207]. In clinical trials, everolimus has shown moderate success in stabilizing disease progression, particularly when used in combination with other therapies [100]. However, mTOR inhibition alone is frequently insufficient owing to pathway resistance mechanisms, such as feedback activation of AKT, leading to efforts to combine mTOR inhibitors with PI3K or AKT inhibitors [208]. Given the complexity of the PI3K/AKT/mTOR pathway and its interaction with other signaling networks, combination therapies are often required to achieve therapeutic efficacy in PDEECs [40]. Dual inhibition with PI3K and mTOR inhibitor agents, such as BEZ235 or BKM120, targets multiple nodes in the pathway, overcoming single-agent resistance and enhancing cell cycle arrest and apoptosis in preclinical models [209]. Since some endometrial cancers express estrogen receptors, combining PI3K/AKT pathway inhibition with hormonal agents (e.g., letrozole) has shown efficacy in hormone receptor-positive cancers [66]. Given the immunosuppressive microenvironment created by the PI3K/AKT/mTOR pathway, combining its inhibitors with immune checkpoint inhibitors (e.g., anti-PD-1/PD-L1) may enhance immune activation against tumors, particularly in cases of high mutational burden or microsatellite instability [210].

Despite the promise of targeting the PI3K/AKT/mTOR pathway, drug resistance remains a significant challenge. Mechanisms of resistance include feedback activation of other pathways (e.g., MAPK/ERK), mutations in downstream targets, and pathway crosstalk that compensates for blocked signaling [153]. One of the critical side effects of PI3K/AKT/mTOR pathway inhibitors in the treatment of PDEECs is immune suppression, which can lead to life-threatening infections [204]. These inhibitors, while effective in blocking tumor growth and survival, also impair key immune functions by suppressing T cell activation, inhibiting antigen presentation, and reducing the proliferation of immune cells that are critical for pathogen defense. Specifically, mTOR inhibitors (e.g., everolimus, temsirolimus) are well known to reduce IL-2 production, leading to diminished T-cell and NK-cell responses, thereby increasing the risk of opportunistic infections such as Pneumocystis jirovecii pneumonia (PJP), cytomegalovirus (CMV) reactivation, and invasive fungal infections [204,205,206,207]. Similarly, PI3K inhibitors (e.g., alpelisib and buparlisib) can cause neutropenia and lymphopenia, further compromising the host’s ability to fight infections. In patients with PDEEC, who may already be immunocompromised due to chemotherapy or advanced disease, this immune suppression can result in severe sepsis, multiorgan failure, or fatal respiratory infections [153,209,210]. Given these risks, patients receiving PI3K/AKT/mTOR inhibitors require close monitoring for signs of infection, prophylactic antimicrobials when indicated, and timely management of immune-related complications to prevent life-threatening outcomes [40,210].

To address this, ongoing research has focused on identifying biomarkers predictive of response to specific inhibitors and developing third-generation inhibitors with increased specificity and reduced toxicity [66]. Additionally, a more precise molecular classification of EECs based on comprehensive genomic profiling will aid in selecting patients who may benefit the most from PI3K/AKT/mTOR-targeted therapies. Thus, targeting the PI3K/AKT/mTOR pathway in PDEECs holds significant therapeutic potential due to frequent mutations in pathway components, such as PIK3CA and PTEN [150]. Combining pathway inhibitors with other therapeutic modalities, such as hormonal or immune therapies, may overcome resistance mechanisms and improve clinical outcomes in this aggressive subset of endometrial cancer [211]. Further studies are needed to optimize combination regimens, develop predictive biomarkers, and personalize treatment strategies for patients.

### 7.2. Immune Checkpoint Inhibitors in MSI-High Tumors

Targeting immune checkpoint inhibitors, specifically programmed death-1 (PD-1) and programmed death ligand-1 (PD-L1) inhibitors, has emerged as a highly effective therapeutic approach for MSI-high (MSI-H) tumors, including PDEECs [212]. MSI-H status in endometrial cancers is frequently associated with mismatch repair (MMR) deficiency, leading to a high mutational burden and increased neoantigen presentation [139]. This immunogenic landscape makes MSI-H tumors particularly responsive to immune checkpoint inhibition [213]. In poorly differentiated MSI-H endometrial cancers, MMR deficiency results in frameshift mutations and neoantigens that attract immune cells to the tumor microenvironment [214]. The elevated mutational burden is thought to enhance the visibility of these tumors to the immune system, amplifying the effects of checkpoint inhibition [139]. Consequently, MSI-H endometrial cancers are more responsive to checkpoint blockade therapy than microsatellite-stable (MSS) cancers [125]. Numerous studies have demonstrated the efficacy of immune checkpoint inhibitors in MSI-H and MMR-deficient endometrial cancers [25]. Pembrolizumab, an anti-PD-1 antibody, was granted FDA approval in 2017 for use in any solid tumor with MSI-H or MMR-deficient status, irrespective of tissue origin [213]. Clinical trials, such as KEYNOTE-158, have shown that pembrolizumab can produce durable responses in MSI-H endometrial cancers [139]. Patients with MSI-H PDEECs have shown particularly high response rates, with some achieving complete or partial responses, which are often maintained over extended periods [161].

Some studies have also evaluated combination strategies, such as nivolumab (an anti-PD-1 antibody) and ipilimumab (an anti-CTLA-4 antibody) [215]. This combination has demonstrated improved response rates in MSI-H endometrial cancers, leveraging dual checkpoint blockades to enhance the immune response [4]. Dual inhibition can activate both effector T-cells and inhibit regulatory T-cells, which otherwise suppress immune responses, providing a robust antitumor effect [216]. MSI-H status itself is an important biomarker for response to checkpoint inhibitors. Tumors with MMR deficiency accumulate mutations at higher rates, leading to a more diverse repertoire of neoantigens that are more likely to be recognized by the immune system when immune suppression is lifted by checkpoint inhibitors [217]. Studies have shown that MSI-H status correlates with higher PD-L1 expression in endometrial cancers, further supporting the rationale for targeting the PD-1/PD-L1 axis in these tumors [9]. Other biomarkers under investigation include tumor mutational burden (TMB) and T-cell infiltration. Both these factors are generally higher in MSI-H EECs and are predictive of responsiveness to immune checkpoint inhibitors [218]. Analyzing TMB and immune infiltration patterns could provide additional information for the personalized treatment of PDEECs [219].

While checkpoint inhibitors alone have shown efficacy in MSI-H EECs, combining these agents with other treatments may further enhance therapeutic responses, particularly in poorly differentiated tumors with aggressive behavior [220]. Combining immune checkpoint inhibitors with therapies that target PI3K/AKT/mTOR pathway alterations, frequently found in endometrial cancers, may enhance outcomes by simultaneously addressing tumor proliferation and immune evasion mechanisms [221]. Radiotherapy can cause an immunostimulatory effect through tumor antigen release and may synergize with immune checkpoint blockade. This combination has shown promise in preliminary studies, although more research is required in the context of MSI-H EECs [222]. Some preclinical studies have suggested that epigenetic therapies can modulate the tumor microenvironment to make it more receptive to immune infiltration. Combining immune checkpoint blockade with drugs such as DNA methyltransferase inhibitors could enhance response rates in MSI-H PDEECs [211].

Despite the promising responses observed, not all patients with MSI-H endometrial cancers respond to checkpoint inhibitors, and acquired resistance remains a challenge [223]. Some tumors may lose the ability to present neoantigens due to mutations in the components of the antigen presentation machinery, reducing the efficacy of checkpoint inhibitors [25]. The immunosuppressive tumor microenvironment in PDEECs, influenced by factors such as TGF-β and regulatory T-cells, can hinder effective T-cell infiltration and activation, limiting the efficacy of immune checkpoint blockade [89]. Research is ongoing to identify predictive biomarkers for responsiveness to immune checkpoint inhibitors beyond MSI-H status and to develop strategies to overcome immune resistance [139]. Resistance to immune checkpoint inhibitors in PDEECs arises through multiple mechanisms that enable tumor cells to evade immune detection and suppression [221]. One key mechanism is the loss or downregulation of major histocompatibility complex (MHC) class I expression, which reduces antigen presentation and impairs T-cell recognition [223]. Additionally, tumors may exhibit an immunosuppressive microenvironment characterized by increased regulatory T cells (Tregs), myeloid-derived suppressor cells (MDSCs), and tumor-associated macrophages (TAMs), all of which inhibit effective antitumor immune responses [139]. Genetic and epigenetic alterations, such as PTEN loss or activation of the WNT/β-catenin pathway, can further contribute to immune evasion by decreasing T-cell infiltration and promoting an immune-cold tumor phenotype [25]. Moreover, upregulation of alternative immune checkpoints, such as TIM-3, LAG-3, and TIGIT, can compensate for PD-1/PD-L1 blockade, leading to continued immune resistance [25,211]. Overcoming these challenges may require combination therapies, including dual checkpoint inhibition, immune-modulating agents, or personalized approaches based on tumor-specific immune signatures [211]. Personalized combination therapies that integrate immune checkpoint inhibition with other treatment modalities are a promising area for future research in PDEECs, offering the potential to improve survival and quality of life for patients with these challenging tumors.

### 7.3. PARP Inhibitors in Homologous Recombination Deficiency (HRD) Cases

Targeting PARP inhibitors in cases of homologous recombination deficiency (HRD) has shown promising potential as a therapeutic strategy for various cancers, and there is growing interest in their application in PDEECs [60]. These cancers often exhibit aggressive behavior and resistance to conventional treatments, leading researchers to explore targeted therapies that exploit specific genetic vulnerabilities [224]. As shown in Figure 8, PARP inhibitors are particularly promising for tumors with HRD, a characteristic that often results from mutations in genes such as BRCA1, BRCA2, and other genes critical for DNA damage repair through homologous recombination [60]. PARP inhibitors, such as olaparib, niraparib, and rucaparib, prevent the repair of SSBs by inhibiting PARP activity, leading to the accumulation of DNA damage [60]. In cells with functioning homologous recombination (HR) repair pathways, this accumulated damage can be repaired accurately [60]. However, in HRD tumors, where genes such as BRCA1, BRCA2, or other HR-related genes are mutated, cells lack the ability to perform high-fidelity DNA repair [225]. This inability to repair damaged DNA leads to double-strand breaks (DSBs) during replication which accumulate and eventually induce cell death through a process known as “synthetic lethality” [226].

Therefore, PARP inhibitors selectively kill HRD cells while sparing normal cells, making them ideal for targeting HRD cancers, including PDEECs, with similar deficiencies [227]. Endometrial cancers, particularly poorly differentiated epithelial subtypes, can exhibit mutations that impair DNA repair pathways, including HRD. This subset of PDEECs with HRD presents an opportunity for targeted therapy with PARP inhibitors [28]. Testing for HRD status can involve identifying specific mutations in HR genes (such as BRCA1/2) or measuring genomic instability signatures indicative of HRD, such as loss of heterozygosity (LOH), large-scale state transitions, and telomeric allelic imbalance. These tests help identify patients with endometrial cancer who may benefit from PARP inhibition [60]. Studies on ovarian and breast cancers have shown the strong efficacy of PARP inhibitors in HRD-positive cases, setting a precedent for their potential use in endometrial cancer [228]. Olaparib and niraparib have shown efficacy in HRD-positive cancers across various tumor types. For instance, olaparib has demonstrated improved progression-free survival in HRD-positive cases, an effect also observed in early phase trials involving endometrial cancer [229]. To enhance the effects of PARP inhibitors, combination strategies with agents such as immune checkpoint inhibitors (ICIs) are being explored [228].

The rationale for this combination lies in the immunogenic nature of HRD tumors, which often exhibit high neoantigen loads, potentially making them more responsive to ICIs [60]. Combining PARP inhibitors with ICIs, such as pembrolizumab, has shown synergistic effects in clinical trials for HRD cancers, offering potential benefits in PDEECs [230]. Despite the promise of PARP inhibitors, resistance can emerge in HRD tumors, limiting the duration and effectiveness of treatment [60]. Some tumors develop secondary mutations in HR genes that restore homologous recombination repair, making them less sensitive to PARP inhibitors. Increased drug efflux and changes in the DNA damage response pathway can also contribute to resistance [164]. To overcome these challenges, researchers are investigating combination therapies, including those that pair PARP inhibitors with ATR inhibitors, which target other parts of the DNA repair pathway [60]. Combining PARP inhibitors with drugs targeting cell cycle checkpoint kinases (CHK1 or WEE1) may further enhance therapeutic efficacy in endometrial cancer [60].

Expanding the clinical use of PARP inhibitors to HRD-positive PDEECs could improve outcomes in patients with aggressive tumors [231]. Improved biomarker testing for HRD, including more comprehensive assays that capture HRD-associated genomic instability, can help identify endometrial cancer patients most likely to benefit from PARP inhibition [60]. As our understanding of HRD and synthetic lethality mechanisms grows, personalized approaches can optimize the therapeutic use of PARP inhibitors in HRD-positive EECs [232]. Endometrial cancer-specific trials evaluating PARP inhibitors alone or in combination with other agents will further elucidate their role in managing PDEECs [233]. The use of PARP inhibitors represents a promising therapeutic strategy for treating PDEECs with HRD. These inhibitors exploit synthetic lethality, targeting cells with defective HR repair mechanisms and leading to selective tumor cell death [234]. Combination therapies and ongoing research hold the potential to improve outcomes in HRD-positive endometrial cancer cases. Nevertheless, challenges such as resistance mechanisms remain. Resistance to PARP inhibitors (PARPis) in PDEECs arises through various molecular and cellular mechanisms that restore DNA repair capabilities or bypass the need for PARP-mediated repair [60]. One major mechanism is the restoration of homologous recombination (HR) proficiency, often through secondary mutations in BRCA1, BRCA2, or other HR-related genes that reverse deleterious mutations and restore DNA repair functions [231]. Additionally, loss of 53BP1 or REV7, key factors in the alternative non-homologous end-joining (NHEJ) pathway, can enable cancer cells to repair DNA double-strand breaks independently of HR, reducing sensitivity to PARPis. Increased drug efflux through upregulation of ATP-binding cassette (ABC) transporters, such as ABCB1 (P-glycoprotein), can also lead to reduced intracellular drug accumulation and diminished efficacy [232].

Furthermore, epigenetic reprogramming and alterations in replication fork stability, including increased stabilization of stalled forks, can help tumor cells resist PARP inhibition [230]. Overcoming these resistance mechanisms may require combination strategies, such as co-targeting ATR, CHK1, or immune checkpoint pathways, to enhance PARPi sensitivity and improve treatment outcomes in PDEECs [231]. With further clinical validation, PARP inhibition could become a cornerstone therapy for managing PDEECs, offering a targeted approach that addresses the unique genomic vulnerabilities of these challenging tumors [59].

## 8. Clinical Implications and Personalized Treatment Strategies in PDEECs

PDEECs are characterized by high-grade histopathological features and aggressive clinical behavior, making treatment and prognostication particularly difficult [235]. With advancements in genomic profiling and molecular diagnostics, precision medicine has emerged as a promising approach for the treatment of these tumors, facilitating personalized treatment plans and improving clinical outcomes. Molecular profiling of PDEECs has transformed traditional approaches by offering insights into the genetic landscape of tumors, enabling the selection of therapies that target specific mutations and pathways [236]. The Cancer Genome Atlas (TCGA) has identified four key molecular subtypes of ECs—POLE ultramutated, microsatellite instability-high (MSI-H), copy-number low, and copy-number high—which are crucial for personalizing treatment strategies as they reflect a distinct tumor biology [237]. Molecular profiling enables clinicians to identify actionable mutations. For instance, aberrations in the PI3K/AKT/mTOR pathway, particularly mutations in PIK3CA, are common in PDEECs, making it a potential target for PI3K inhibitors [238]. In MSI-H tumors, immune checkpoint inhibitors, such as pembrolizumab, have shown promise owing to the high mutational burden and presence of neoantigens, which make these tumors more susceptible to immune-mediated destruction [239]. Although precision medicine offers hope for personalized therapies, several challenges remain in the treatment of PDEECs. PDEECs exhibit significant genetic and phenotypic heterogeneity both within the primary tumor and between primary and metastatic sites [238]. This variability complicates the identification of uniform molecular targets and contributes to our understanding of resistance mechanisms. For example, certain regions of the tumor might harbor actionable mutations, while others do not, limiting the efficacy of targeted therapies [237].

Resistance to targeted therapies often arises from secondary mutations or compensatory signaling pathways. For example, while PI3K inhibitors may initially show efficacy, tumors can activate alternative pathways, such as MAPK, leading to resistance [240]. Resistance to PI3K inhibitors in PDEECs arises through multiple mechanisms, including compensatory activation of alternative signaling pathways, genetic alterations, and adaptive tumor microenvironment changes [238]. One major resistance mechanism involves the upregulation of parallel pathways, such as the MAPK/ERK and AKT/mTOR signaling cascades, which can bypass PI3K inhibition and sustain tumor proliferation and survival [237]. Additionally, genetic mutations in PIK3CA, loss of PTEN, and activating alterations in KRAS or FGFR2 contribute to intrinsic or acquired resistance. Epigenetic modifications and metabolic reprogramming further enhance the adaptability of PDEECs, thereby reducing the efficacy of PI3K-targeted therapies [239,240]. Moreover, the tumor microenvironment, including immune suppression and stromal interactions, fosters resistance by modulating cytokine signaling and promoting a pro-survival niche [240]. Overcoming these resistance mechanisms may require combination strategies, such as dual-pathway inhibition or immune checkpoint blockade, to enhance the therapeutic efficacy in PDEECs [237].

The dynamic nature of these tumors underscores the need for combination therapies or sequential treatment regimens to overcome resistance. In many PDEECs, it can be difficult to identify actionable mutations [56]. Even with comprehensive genomic profiling, some tumors lack clear molecular targets, limiting the scope of precision medicine. In such cases, patients may be treated with standard chemotherapy or radiation, which often has limited success in this high-risk population [240]. The integration of histopathology with molecular genomics represents the future of personalized cancer care for PDEECs. Traditionally, PDEECs have been classified and treated based on histopathological characteristics [241]. However, molecular profiling offers a more nuanced understanding of tumor biology. By integrating these approaches, clinicians can improve the accuracy of prognostication and refine treatment selection [29]. For example, the identification of specific mutations or molecular subtypes can complement histopathological assessment and provide a more comprehensive picture of tumor behavior [242]. As our understanding of the molecular landscape of PDEECs deepens, biomarker-driven clinical trials will become increasingly important [8]. Trials that stratify patients based on specific genetic alterations (such as PIK3CA mutations or MSI status) can help identify the most effective targeted therapies in these populations. Moreover, the development of multi-gene panels for routine clinical use could facilitate rapid and cost-effective molecular profiling, making precision medicine more accessible [60].

The future of personalized treatment for PDEECs likely lies in combination therapies that target multiple pathways simultaneously or sequentially. For example, combining PI3K inhibitors with inhibitors of other pathways (such as MAPK or Wnt) may prevent or delay the development of resistance [243]. Similarly, combining targeted therapies with immunotherapies holds promise, particularly in tumors that exhibit both actionable mutations and a high mutational burden [60]. Advances in liquid biopsy technologies offer a non-invasive means of monitoring tumor evolution and detecting resistance mutations in real time [244]. For PDEECs, in which tumor heterogeneity is a significant challenge, liquid biopsies could provide a more comprehensive view of the genetic landscape of the tumor, enabling more precise treatment adjustments over the course of therapy [101]. Liquid biopsy is emerging as a minimally invasive tool for real-time tumor monitoring and treatment response assessment in PDEECs [9,14]. These tests analyze circulating tumor DNA (ctDNA), circulating tumor cells (CTCs), and other biomarkers present in the blood, providing valuable insights into tumor burden, genetic evolution, and therapy resistance without the need for repeated tissue biopsies [90].

In PDEECs, where tumors are highly aggressive and prone to metastasis, liquid biopsies can detect minimal residual disease (MRD), identify actionable mutations (e.g., PIK3CA, TP53, and PTEN), and guide targeted therapy selection [101]. Additionally, serial monitoring of ctDNA levels allows clinicians to track treatment efficacy, detect early signs of recurrence, and adjust therapy before clinical progression occurs [100]. However, challenges such as low ctDNA levels in early stage disease, standardization of assays, and sensitivity issues remain. Despite these limitations, liquid biopsies have the potential to revolutionize precision oncology in PDEECs by enabling dynamic and patient-specific treatment strategies to improve outcomes [24,102]. The transition from histomorphology to molecular profiling of PDEECs has opened new avenues for personalized treatment [245]. While precision medicine has shown promise, challenges such as tumor heterogeneity, resistance mechanisms, and difficulty in identifying actionable mutations highlight the need for ongoing research [60]. Future directions include the integration of molecular and histopathological data, biomarker-driven trials, and innovative combinatorial strategies to improve patient outcomes in this aggressive subset of endometrial cancer.

## 9. Model Systems for Studying PDEECs

The study of PDEECs requires reliable preclinical models that accurately recapitulate the molecular, histological, and clinical features of these aggressive tumors. Various in vitro (2D and 3D models), in vivo (patient-derived xenograft (PDX) and genetically engineered mouse model (GEMM)), and ex vivo (tumor slice cultures) model systems are used to investigate tumor biology, drug responses, metastasis, and immune interactions in PDEECs [58]. The selection of an appropriate model depends on the research objective, with PDXs being ideal for drug testing, GEMMs for molecular studies, and organoids for precision medicine applications [59]. Two-dimensional cell line models are widely used to study PDEECs, providing a convenient and reproducible system for investigating tumor biology, drug responses, and molecular signaling pathways [25]. Several high-grade endometrial cancer cell lines have been developed to model different subtypes of PDEECs. Notable examples include ARK1 and ARK2, which resemble high-grade serous-like endometrial carcinoma, and Hec50 and SPAC1-L, which represent poorly differentiated endometrioid carcinomas [25,59]. These cell lines are valuable tools for exploring key oncogenic pathways, such as PI3K/AKT/mTOR signaling, studying cell proliferation and apoptosis, and assessing chemotherapy resistance mechanisms [58]. Despite their utility, 2D models have significant limitations, primarily due to the lack of a tumor microenvironment, stromal interactions, and cellular heterogeneity, which reduces their clinical relevance [25]. As a result, while 2D cell lines provide initial insights into PDEEC pathophysiology, more advanced models, such as 3D organoids and patient-derived xenografts (PDXs), are increasingly being used to enhance translational research [58,59].

Patient-derived tumor organoids (PDTOs) represent an advanced three-dimensional (3D) culture system that more accurately models PDEECs than traditional 2D cell lines [205]. These self-organizing structures are derived from primary tumor tissues, allowing them to retain the key features of the original cancer, including tumor architecture, genetic heterogeneity, and patient-specific drug response patterns [206]. Owing to their physiological relevance, organoid models are particularly useful for studying targeted therapies, such as PI3K/AKT/mTOR inhibitors and immune checkpoint inhibitors, providing a powerful platform for precision oncology and personalized medicine. Furthermore, they facilitate research into tumor–stroma interactions, therapy resistance, and metastatic potential [150]. Despite these advantages, PDTOs have significant limitations, including high costs, complex culture conditions, and the technical expertise required for their establishment and maintenance [25]. Nevertheless, as organoid technology advances, it is expected to play a critical role in preclinical drug testing and biomarker discovery in PDEECs [205].

Patient-derived xenograft (PDX) models are essential in vivo systems for studying PDEECs. In this approach, tumor tissue from PDEEC patients is implanted either subcutaneously or orthotopically (in the uterus) into immunocompromised mice, such as NSG (NOD scid gamma), SCID (severe combined immunodeficiency), or nude mice [190]. These models effectively retain the genetic, epigenetic, and histological characteristics of the original tumor, making them highly valuable for drug efficacy testing, biomarker discovery, and investigating mechanisms of therapy resistance [95]. By closely mimicking human tumor biology, PDX models enable the preclinical assessment of novel therapeutics, including PI3K/AKT/mTOR pathway inhibitors and chemotherapy regimens [124]. However, a major limitation of PDX models is the absence of a functional immune system in host mice, which prevents the study of immune checkpoint inhibitors and other immunotherapies [150]. Despite this, humanized PDX models, where human immune cells are reconstituted in mice, are being explored to overcome this challenge and expand the use of PDXs in immuno-oncology research [209].

Genetically engineered mouse models (GEMMs) provide a powerful platform for studying the molecular mechanisms driving PDEECs by introducing endometrial-specific genetic alterations that mimic those observed in human tumors. Pik3ca, PTEN, and TP53 mutant mice have been widely used to investigate how dysregulation of PI3K/AKT/mTOR signaling, loss of tumor suppression, and genomic instability contribute to PDEEC progression [190,209]. A notable example is the Lkb1fl/fl Ptenfl/fl mouse model, which develops high-grade endometrial carcinomas resembling PDEECs, making it an effective system for studying tumor initiation, metastasis, and therapeutic response [150]. Unlike patient-derived xenografts (PDXs), GEMMs retain a functional immune system, making them particularly valuable for immunotherapy research, including studies on immune checkpoint inhibitors [210]. However, GEMMs are resource-intensive and require long development times, high costs, and complex breeding strategies, which can limit their widespread use [95]. Despite these challenges, GEMMs remain a critical tool for investigating tumor biology, drug resistance, and novel treatment strategies for PDEECs [150].

Ex vivo and computational models provide innovative approaches for studying PDEECs by enabling personalized drug testing and predictive modeling of tumor behavior [66]. Tumor slice cultures involve maintaining fresh PDEEC tumor slices in culture while preserving the original stromal, immune, and tumor microenvironment components [150]. This system allows personalized therapy testing on patient-derived tissues, helping to evaluate drug sensitivity and resistance in a clinically relevant setting [9]. However, a major limitation of tumor slice cultures is their short lifespan, which restricts long-term studies [25]. In contrast, computational and AI-based models use in silico machine learning algorithms to predict mutation-driven pathways, drug responses, and survival outcomes in PDEECs [209]. These models analyze large datasets to identify potential therapeutic targets and optimize treatment selection, making them highly valuable for precision medicine approaches [25]. While ex vivo models provide functional validation, computational models offer rapid and scalable predictions, and together they enhance our ability to develop more effective and individualized treatments for patients with PDEEC [66,150].

## 10. Surgical Interventions for PDEECs

Surgical intervention is the primary treatment modality for PDEECs, given their aggressive nature and high risk of extrauterine spread. The extent of surgery depends on the tumor stage, histologic subtype, and patient factors, such as age, comorbidities, and performance status [246]. Standard surgical management aims for complete tumor resection, accurate staging, and cytoreduction when necessary [247]. Total hysterectomy with bilateral salpingo-oophorectomy (TH + BSO) is the standard surgical procedure for the treatment of PDEECs and other endometrial malignancies, regardless of the disease stage [246,247,248]. The procedure involves complete removal of the uterus, cervix, fallopian tubes, and ovaries, ensuring the elimination of the primary tumor and reducing the risk of disease recurrence [248,249]. Given that PDEECs are high-grade aggressive tumors with a high propensity for extrauterine spread, removal of both ovaries (oophorectomy) is essential, even in postmenopausal patients [249]. Although many endometrial cancers are estrogen-dependent, PDEECs are often estrogen-independent, indicating that ovarian removal is necessary not only for hormonal suppression but also to prevent micrometastatic disease or direct ovarian involvement. By excising all potential sites of tumor persistence, TH + BSO remains a cornerstone of surgical management, frequently followed by adjuvant therapy to improve long-term outcomes [246,247,248,249].

Sentinel lymph node (SLN) mapping and systematic lymphadenectomy are two key approaches for lymph node assessment in PDEECs, helping determine disease spread and prognosis [250]. SLN biopsy is primarily used in early-stage cases (FIGO stage I-II) to detect lymphatic dissemination while minimizing surgical morbidity. This technique involves the injection of indocyanine green (ICG) or technetium-99m, which allows surgeons to identify and remove the first-draining lymph nodes most likely to contain metastases [251]. In contrast, systematic pelvic and para-aortic lymphadenectomy is recommended for high-risk or advanced PDEECs where nodal involvement is suspected. This procedure involves the removal of multiple lymph nodes in the pelvic and para-aortic regions to determine the extent of the disease [250,251,252]. Given that PDEECs have a high propensity for lymphatic spread (~30–50% in advanced cases), accurate nodal staging is critical for determining prognosis and guiding adjuvant treatment decisions, such as the use of chemotherapy or radiotherapy [253]. While SLN mapping is a less invasive approach with fewer complications, systematic lymphadenectomy provides comprehensive disease evaluation in cases with a high risk of metastasis [254].

Omentectomy is a surgical procedure involving the removal of the omentum, a fatty tissue layer that drapes over the abdominal organs. This procedure is particularly important in the management of high-grade serous and undifferentiated carcinomas, which have a strong propensity for peritoneal dissemination [255]. Given that PDEECs, especially serous and clear cell subtypes, frequently spread beyond the uterus and seed the peritoneal cavity, omentectomy serves both therapeutic and staging roles [256]. It is indicated in cases where peritoneal involvement is suspected, either based on preoperative imaging or intraoperative findings of tumor spread [257]. Omentectomy helps remove microscopic metastatic disease, improves accurate cancer staging, and informs decisions regarding adjuvant therapy, such as chemotherapy [255,256,257]. In advanced cases, it is often performed as part of cytoreductive surgery to achieve optimal tumor debulking and improve overall survival outcomes [258].

Cytoreductive surgery (debulking surgery) is a critical procedure in the management of advanced-stage PDEECs (FIGO stage III-IV), aiming to remove all visible tumor deposits within the abdominal and pelvic cavities [259]. The primary goal of this surgery is to achieve optimal debulking, defined as a residual tumor < 1 cm, which has been associated with significantly improved survival outcomes [259,260]. Given the aggressive nature of PDEECs and their high propensity for peritoneal and lymphatic spread, cytoreduction plays a crucial role in reducing tumor burden and enhancing the efficacy of adjuvant chemotherapy [261]. Studies have demonstrated that patients undergoing complete or near-complete tumor resection experience longer progression-free survival (PFS) and overall survival (OS) than those with residual bulky disease [259]. This approach is particularly beneficial for cases of peritoneal carcinomatosis, omental metastases, or nodal involvement, where systemic therapy alone may be insufficient [260]. Although cytoreductive surgery is a high-risk, complex procedure, its role in improving treatment response and long-term outcomes makes it an essential component of multidisciplinary cancer management [262].

Minimally Invasive Surgery (MIS), including laparoscopy or robotic surgery, is typically recommended for early-stage PDEECs in stages I-II, where there is no extrauterine spread. This approach is associated with several benefits, such as reduced blood loss, shorter hospital stays, and faster recovery times compared to open surgery [263]. On the other hand, open surgery or laparotomy is generally preferred for more advanced cases, such as high-grade, bulky tumors, or when there is suspicion of peritoneal metastasis [264]. Laparotomy allows for a more extensive procedure, including lymphadenectomy, omentectomy, and optimal cytoreduction, which are crucial for treating advanced-stage PDEECs [265]. Both surgical approaches have specific indications depending on tumor characteristics and stage, with MIS offering a less invasive option for early-stage cancers and open surgery being essential for more complex cases [266].

Fertility-sparing surgery is a rare and highly selective approach considered only for young patients with grade 1 endometrioid histology, provided there is no myometrial invasion or extrauterine spread [267]. Owing to the aggressive nature of PDEECs, this approach is not recommended for such cases. The procedure typically involves hysteroscopic resection of the tumor followed by progestin therapy, such as megestrol acetate or a levonorgestrel intrauterine device (IUD), to suppress tumor growth [268]. However, fertility-sparing management has significant limitations, including a high recurrence rate, necessitating rigorous follow-up [267]. Given the risk of disease progression, patients who undergo this treatment must be closely monitored, and early definitive hysterectomy is strongly advised after childbearing to ensure long-term oncological safety [267,268].

Surgery plays a crucial role in the management of recurrent PDEECs, with the approach depending on the extent and location of recurrence [269]. Secondly, cytoreductive surgery is considered in selected patients with isolated pelvic or peritoneal recurrence, where complete resection is feasible and may offer improved survival outcomes. In these cases, achieving no residual disease is the primary goal, as it is associated with a better prognosis [262]. However, in patients with widespread recurrence or inoperable disease, palliative surgery may be performed to alleviate symptoms and enhance quality of life. This includes procedures to manage complications, such as bowel obstruction, vaginal bleeding, or ureteric obstruction, which can significantly impact patient well-being [259]. While surgery remains an important tool in recurrent PDEEC management, careful patient selection is essential to balance the potential benefits with surgical risks, often in conjunction with systemic therapies, such as chemotherapy, hormonal therapy, or radiation, for optimal disease control [258].

## 11. Conclusions

Exploration of the genomic landscape of PDEECs reveals complex and heterogeneous tumor biology. Key genetic alterations frequently observed in these cancers include mutations in TP53, PIK3CA, PTEN, ARID1A, and CTNNB1, as well as significant copy number alterations and epigenetic modifications. These molecular insights help to classify PDEECs into distinct subgroups, including those with MSI-H and POLE-ultramutated tumors, each with its own implications for prognosis and therapeutic responsiveness. The identification of critical pathways, such as PI3K/AKT/mTOR and the p53 pathway, has provided a framework for developing targeted therapies, especially in aggressive cases where traditional histopathology alone fails to guide effective treatment. The integration of molecular profiling into the clinical management of PDEECs is increasingly recognized as essential. Historically, these tumors have been difficult to classify and treat based solely on histomorphology due to their poor differentiation and aggressive nature. Molecular data not only enhance diagnostic precision but also enable the stratification of patients into subgroups with distinct prognostic and therapeutic needs. For instance, tumors with MSI-H status may benefit from immune checkpoint inhibitors, whereas those harboring PIK3CA mutations may be targeted using PI3K pathway inhibitors. Molecular profiling allows for more personalized treatment strategies, offering hope for improved outcomes in a population that has historically faced poor prognosis.

Despite significant progress, there is a need for more comprehensive studies that integrate histomorphology, genomics, and therapeutic responses. These studies would provide greater insight into how molecular alterations interact with histopathological features to drive tumor behavior and therapy resistance. Longitudinal studies and real-time monitoring, perhaps through the use of liquid biopsies, could also help track tumor evolution and guide adaptive treatment strategies. Furthermore, ongoing clinical trials targeting the key pathways identified in these genomic studies are likely to refine therapeutic approaches for PDEECs. Ultimately, the combination of genomic profiling and histopathological assessment will play a pivotal role in improving prognostication and patient outcomes, particularly as new biomarkers and molecular targets are validated for clinical use.

## Figures and Tables

**Figure 1 cells-14-00382-f001:**
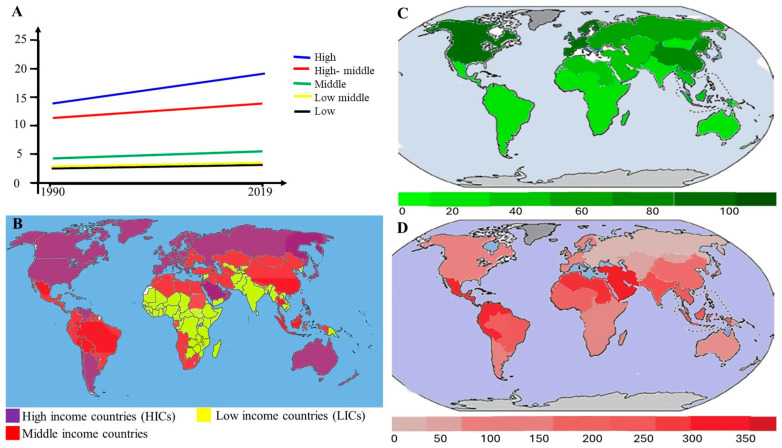
Global incidence and mortality rates of endometrial cancer. (**A**) The graph depicts the global trends in ASIR for endometrial cancer over the last 30 years, from 1990 to 2019, in countries grouped by economic status. The figure shows an increase in the incidence of the disease, regardless of income. However, a trend is visible in that the higher the income of a country, the greater the incidence of endometrial cancer. The study of the global burden of disease classifies different regions of the world. (**B**) Map depicting high-, middle-, and low-income countries. (**C**) This figure shows the age-standardized incidence rate per 100,000 in 2019 for each of the adjacent GBD regions. High-income North America, Western Europe, and East Asia are the GBD regions with the highest incidence. GBD values can be used to calculate changes in the incidence of disease over time, represented as a percentage. (**D**) Shows the global increase in endometrial cancer incidence 1990–2019. This was calculated by dividing the ASIR in 2019 by the ASIR from 1990 and multiplying by 100. The map and heatmap indicate that regions associated with lower income have faster-growing rates of endometrial cancer incidence. The increase in the incidence in low-to middle-income regions is thought to be linked to changes in lifestyle and a higher prevalence of risk factors such as obesity and inactivity levels. ASIR: age-standardized incidence rate. SDI: socio-demographic index. GBD: Global Burden of Disease (Figure adapted from [3]).

**Figure 2 cells-14-00382-f002:**
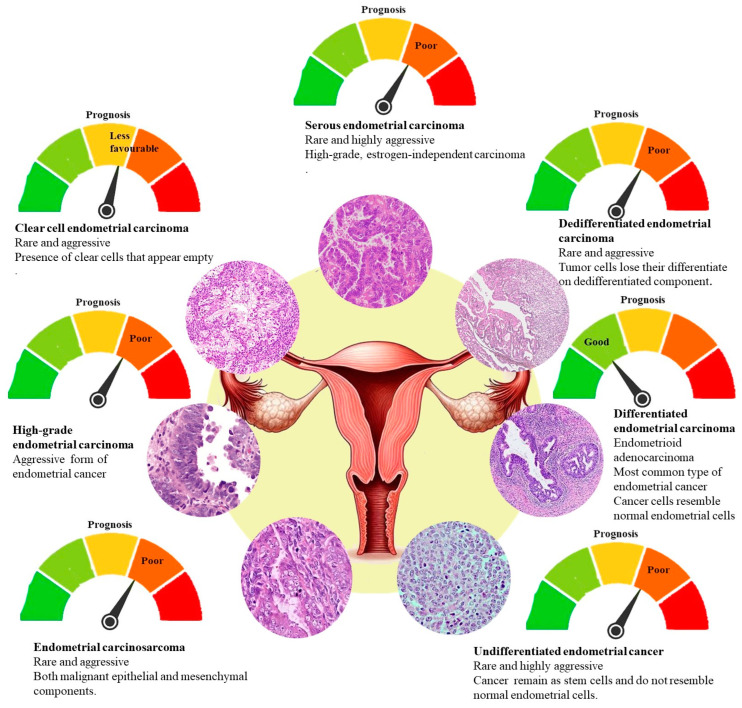
The histologic subtypes of poorly differentiated epithelial endometrial cancers (PDEECs). The histological subtypes of PDEECs are a heterogeneous group of aggressive endometrial malignancies characterized by high-grade cellular atypia, increased mitotic activity, and poor glandular differentiation. The subtypes depicted include serous carcinoma, high-grade endometrioid carcinoma, clear cell endometrial carcinoma, dedifferentiated endometrial carcinoma, differentiated endometrial carcinoma, and undifferentiated endometrial carcinoma. Each histological subtype is associated with distinct molecular alterations, prognostic implications, and treatment responses. Representative histological images stained with hematoxylin and eosin (H&E) highlight the morphological diversity among PDEECs. The immunohistochemical markers used for differential diagnosis include p53, Napsin A, and E-cadherin.

**Figure 3 cells-14-00382-f003:**
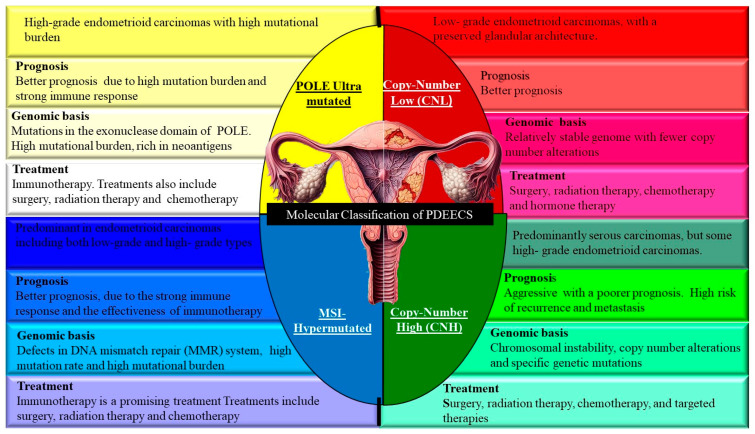
Molecular classification of poorly differentiated epithelial endometrial cancers (PDEECs). This classification provides insights into the biological diversity and clinical behavior of these aggressive tumors. Primarily defined through genomic and transcriptomic analyses, this classification has helped refine the diagnostic criteria, predict prognosis, and identify potential therapeutic targets. It divides PDEECs into four main subtypes, namely, POLE-ultramutated, microsatellite instability-high (MSI-H), copy number low (CNL), and copy number high (CNH), based on specific molecular markers.

**Figure 6 cells-14-00382-f006:**
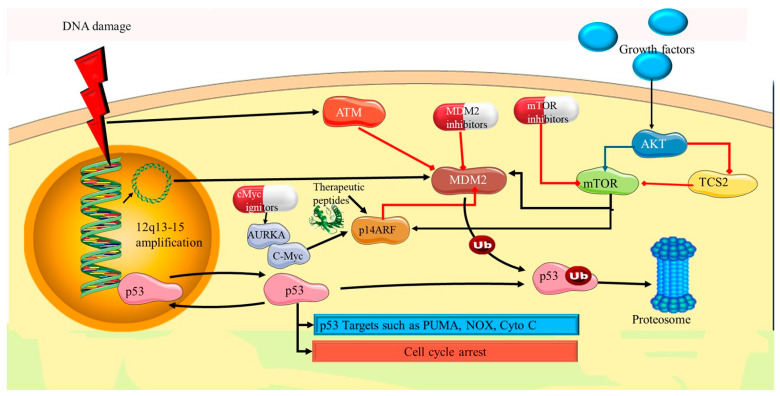
Overview of p53 pathway dysfunction. Dysfunction of the p53 pathway in PDEECs is a driving factor in tumor progression and therapy resistance. Understanding and targeting this dysfunction is central to improving treatment strategies for high-grade aggressive cancers. As research advances, the development of p53-targeted therapies, synthetic lethal approaches, and combination regimens holds promise for addressing the challenges associated with p53-mutant PDEECs (figure adapted from [159]).

**Figure 7 cells-14-00382-f007:**
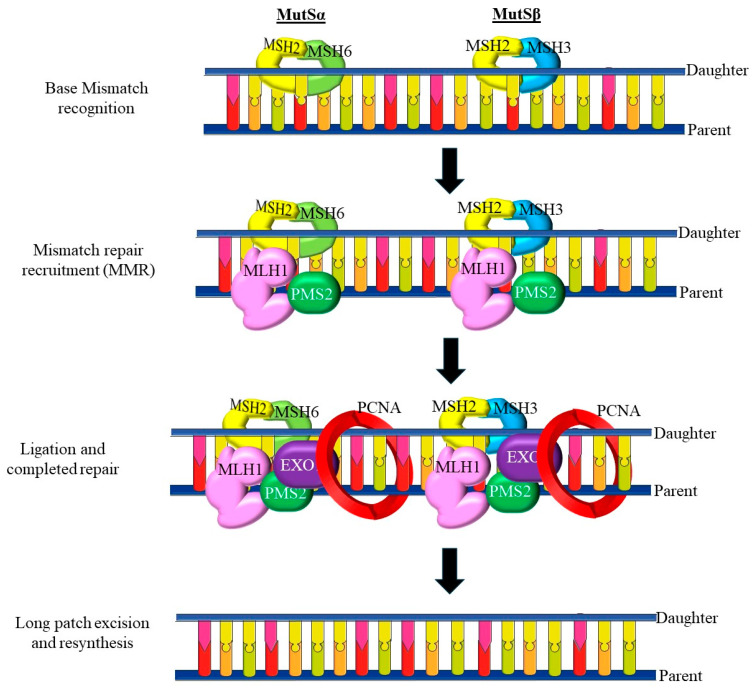
Mismatch repair (MMR) pathway and mechanism. MMR deficiency is a critical factor in the pathology and treatment of PDEECs. By contributing to genomic instability and high MSI, it not only promotes tumor progression but also presents an opportunity for targeted immunotherapy, which could enhance survival in affected patients. Identifying and understanding MMR deficiency allows clinicians to personalize treatment and may pave the way for novel therapeutic strategies for endometrial cancer (figure adapted from [163]).

**Figure 8 cells-14-00382-f008:**
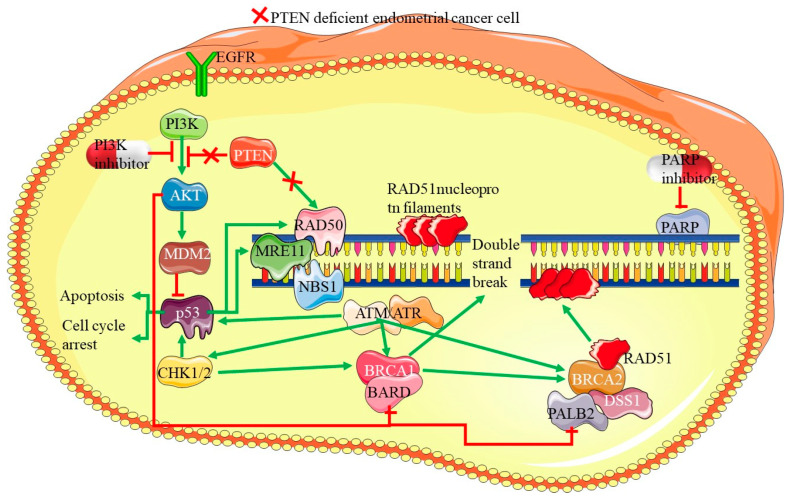
PARP inhibitors in homologous recombination deficiency (HRD) in PDEECs. PARP inhibitors have emerged as a promising therapeutic strategy for treating PDEECs with homologous recombination deficiency (HRD). HRD is a condition in which cells lose the ability to accurately repair double-strand DNA breaks via the homologous recombination (HR) pathway, often due to mutations or deficiencies in key genes, such as BRCA1, BRCA2, and PTEN. HRD makes cancer cells more reliant on alternative DNA repair mechanisms, such as those mediated by PARP. By inhibiting PARP, these drugs trap cancer cells through a cycle of DNA damage, ultimately leading to cell death. In PDEECs, PARP inhibitors show promise as a targeted treatment, particularly for subtypes that are resistant to conventional therapies (figure adapted from [60]).

**Table 1 cells-14-00382-t001:** Histological subtypes of poorly differentiated epithelial endometrial cancers (PDEECs).

PDEEC Subtypes	Description	Reference
High-grade endometrioid carcinoma	Displays solid growth with focal glandular differentiation. Tumor cells exhibit marked nuclear pleomorphism, frequent mitotic figures, and areas of necrosis. Distinguished from low-grade endometrioid carcinoma by the absence of well-formed glands and the presence of diffuse atypia.	[2]
Serous carcinoma	Composed of highly pleomorphic cells arranged in papillary, micropapillary, or solid patterns. Characterized by prominent nucleoli, high mitotic index, and frequent psammoma bodies. Often associated with TP53 mutations and extensive lymphovascular invasion.	[15]
Clear cell carcinoma	Features polygonal or hobnail cells with clear or eosinophilic cytoplasm. Growth patterns include solid, papillary, and tubulocystic structures. Tumor cells frequently express Napsin A and HNF1β.	[12]
Undifferentiated carcinoma	Composed of sheets of highly atypical cells lacking glandular differentiation. Frequent loss of epithelial markers such as E-cadherin, leading to growth as a result of the loss of cohesion. Often diagnosed in association with dedifferentiated carcinoma.	[7]
Dedifferentiated carcinoma	Contains a biphasic pattern with a high-grade endometrioid component adjacent to an undifferentiated carcinoma. Demonstrates loss of clonal differentiation markers, such as SWI/SNF complex proteins (ARID1A, SMARCA4).	[9]

**Table 2 cells-14-00382-t002:** Summary of PTEN mutation types and their effects.

Mutation Type	Examples	Mechanism	Functional Impact
Missense mutations (hotspot mutations)	R130G/Q, C124S, H123Y, G129E	Disrupts the phosphatase domain, impairing enzymatic activity	Partial loss of function, protein may still be detectable on IHC
Frameshift/Nonsense mutations (truncating mutations)	R233*, Y68fs, R335*	Introduces premature stop codons leading to nonsense-mediated decay	Complete loss of PTEN expression, strong PI3K/AKT activation
Large deletions/Copy number loss	LOH at 10q23	Entire gene or large regions deleted	Total absence of PTEN protein, highly aggressive tumor phenotype

**Table 3 cells-14-00382-t003:** Downstream signaling effects of PIK3CA mutations.

Component	Activated by PI3K Mutation	Pro-Tumorigenic Effects	References
AKT (protein kinase B)	Phosphorylated at T308 (by PDK1) and S473 (by mTORC2)	Enhances cell survival, proliferation, metabolism	[65,67]
mTOR (mechanistic target of rapamycin)	Activated downstream of AKT	Promotes protein synthesis, cell growth, metabolic reprogramming	[10,63]
FOXO transcription factors	Inhibited by AKT phosphorylation	Prevents apoptosis, supports immune evasion	[7,65]
BAD (pro-apoptotic protein)	Inactivated by AKT phosphorylation	Suppresses apoptosis, enhances chemoresistance	[64,67]
GSK3β (glycogen synthase kinase 3 beta)	Inhibited by AKT phosphorylation	Deregulates β-catenin signaling, promotes metastasis	[64,69]

## Data Availability

No new data were created or analyzed in this study.

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
