# Peer review of "The Histomorphology to Molecular Transition: Exploring the Genomic Landscape of Poorly Differentiated Epithelial Endometrial Cancers"

_cells, 2025, doi:10.3390/cells14050382_

Round 1

Reviewer 1 Report

Comments and Suggestions for Authors

The provided review article comprehensively discusses poorly differentiated epithelial endometrial cancers. The review covers various aspects of these aggressive malignancies, from their morphological features to their molecular underpinnings and potential therapeutic strategies. Here's an assessment of the writing and potential gaps:

Minor comments:

1)  If authors can provide a simple diagram for Figure 1, just showing Low, high, and middle will be helpful rather than sub-classification with a timeline marking every decade and associated changes as a graph.

2) Figure 2 lacks clarity and will be helpful to provide clarity as it is unreadable.

3)  Can the authors clarify why it is not mentioned and that a proper classification must be included as a figure in the table or elsewhere? As it is not clear what terminology has been used.

2) Figure 4 POLE exonuclease domain mutations are mentioned but not included in the text.

3) The percentages presented in Table 2 for TP53 mutations are reported at 57.7-92%. In comparison, they are reported as being found in up to 90% of serous endometrial carcinomas. A proper classification figure, as mentioned above, will also help provide accurate definitions of the subtypes.

4) Table 2—Separating genetic and epigenetic mutations will help define what is known in the field, which authors fail to mention. Maybe including top epigenetic regulator candidates will be helpful.

Major Comments:

1)  The authors have listed many top mutations and relevant discussions. However, this makes the review word-heavy; trimming the mutations aspect of the sections will make it flow seamlessly. This will help summarise the key points and events to be considered for the future. Maybe summarizing the write-up in the text and creating a table with roles and functions will clarify it and shorten the text or summarize wherever appropriate.

2) While the review mentions resistance to targeted therapies, it could benefit from a more detailed discussion of the specific resistance mechanisms to PI3K inhibitors, immune checkpoint inhibitors, and PARP inhibitors and potential strategies for overcoming these mechanisms.

3) The review touches on epigenetic modifications and their therapeutic potential; it could be valuable to expand on specific epigenetic drugs, their mechanisms, and their clinical trial data.

4) While the review touches on the role of the tumor microenvironment, particularly in immunotherapy, more insights into its complex interactions could be included.

5) The review could be enhanced by discussing the role of liquid biopsies in tumor monitoring and therapy response.

6) While the review references clinical trials, incorporating a table will be helpful. It will clarify the write-up regarding their outcomes, especially regarding targeted therapies and immunotherapies, and strengthen the discussion.

7) The review could benefit from a section on more near-term clinical applications of their understanding of the molecular findings for molecular testing and treatment decisions.

8) Additionally, the authors should summarize what is known in the field of epigenetics and how it should be integrated to address the key challenges. One paragraph summarizing will be beneficial.

Author Response

Reviewer 1

The provided review article comprehensively discusses poorly differentiated epithelial endometrial cancers. The review covers various aspects of these aggressive malignancies, from their morphological features to their molecular underpinnings and potential therapeutic strategies. Here's an assessment of the writing and potential gaps:

Minor comments:

1)  If authors can provide a simple diagram for Figure 1, just showing Low, high, and middle will be helpful rather than sub-classification with a timeline marking every decade and associated changes as a graph.

Response: Thank you for your insightful comments. We have altered figure 1, however we have not simplified the figure as a whole, simpler clearer maps have been used and a map identifying what is meant by low, middle and high income countries, but we have retained the graph shoeing the progression of the incidence of the disease. While we understand the appeal of a simpler diagram, the current figure design serves several important purposes.

  1. It illustrates both broad trends (low, middle, and high) and more nuanced changes within these categories over time.
  2. The timeline marking every decade provides the context for significant changes in incidence rates.
  3. The sub-classification helps to demonstrate the complexity of the disease burden across different regions and socioeconomic groups.

2) Figure 2 lacks clarity and will be helpful to provide clarity as it is unreadable.

Response: We acknowledge this and have modified the text accordingly. The revised section now provides more explanation of the distinct molecular alterations and prognostic implications of the histologic subtypes. We have provided a clearer figure 2

3)  Can the authors clarify why it is not mentioned and why a proper classification must be included as a figure in the table or elsewhere? It is not clear what terminology has been used.

Response: We thank the reviewer for pointing this out. We have incorporated a table (Table 1) to provide clarity on this aspect.

2) Figure 4 POLE exonuclease domain mutations are mentioned but not included in the text. 

Response: We acknowledge this point and have incorporated additional text accordingly.

3) The percentages presented in Table 2 for TP53 mutations are reported at 57.7-92%. In comparison, they are reported as being found in up to 90% of serous endometrial carcinomas. A proper classification figure, as mentioned above, will also help provide accurate definitions of the subtypes.

Response: We appreciate this insightful comment. To clarify, we have added a classification table, as mentioned above, with accurate definitions of the subtypes. This modification has improved the clarity and accuracy of our findings.

4) Table 2—Separating genetic and epigenetic mutations will help define what is known in the field, which authors fail to mention. Maybe including top epigenetic regulator candidates may/will be helpful.

Response: We acknowledge this and have modified the table accordingly. The revised section now distinguishes between genetic and epigenetic mutations.

Major Comments:

1)  The authors have listed many top mutations and relevant discussions. However, this makes the review word-heavy; trimming the mutations aspect of the sections will make it flow seamlessly. This will help summarise the key points and events to be considered for the future. Maybe summarizing the write-up in the text and creating a table with roles and functions will clarify it and shorten the text or summarize wherever appropriate.

Response: We appreciate this insightful comment. To clarify, we have created tables with roles and functions to clarify, shorten and summarize wherever appropriate. This modification has improved the clarity and accuracy of the text.

2) While the review mentions resistance to targeted therapies, it could benefit from a more detailed discussion of the specific resistance mechanisms to PI3K inhibitors, immune checkpoint inhibitors, and PARP inhibitors and potential strategies for overcoming these mechanisms.

Response: Thank you for your valuable suggestion. We have addressed this by incorporating additional sections discussing the specific resistance mechanisms to PI3K inhibitors, immune checkpoint inhibitors, and PARP inhibitors and potential strategies for overcoming these mechanisms in PDEECs.

3) The review touches on epigenetic modifications and their therapeutic potential; it could be valuable to expand on specific epigenetic drugs, their mechanisms, and their clinical trial data.

Response: We acknowledge this point and have modified the text accordingly. The revised section now discusses specific epigenetic drugs, their mechanisms, and clinical trial examples.

4) While the review touches on the role of the tumour microenvironment, particularly in immunotherapy, more insights into its complex interactions could be included.

Response: Thank you for this valuable suggestion. We have addressed the role of the tumor microenvironment, particularly in immunotherapy, and provided more insights into its complex interactions in the manuscript.

5) The review could be enhanced by discussing the role of liquid biopsies in tumour monitoring and therapy response.

Response: We appreciate this insightful comment. To clarify, we have revised and discussed the role of liquid biopsies in tumor monitoring and therapy response. This modification has improved the clarity and accuracy of our manuscript.

6) While the review references clinical trials, incorporating a table will be helpful. It will clarify the write-up regarding their outcomes, especially regarding targeted therapies and immunotherapies, and strengthen the discussion.

Response: Thank you for this valuable suggestion. We have addressed this by referencing specific clinical trials and their outcomes regarding targeted therapies and immunotherapies and have strengthened the discussion sections of the manuscript.

7) The review could benefit from a section on more near-term clinical applications of their understanding of the molecular findings for molecular testing and treatment decisions.

Response: We acknowledge this point and have incorporated examples of near-term clinical applications of molecular findings for molecular testing and treatment decisions.

8) Additionally, the authors should summarize what is known in the field of epigenetics and how it should be integrated to address the key challenges. One paragraph summarizing will be beneficial.

Response: Thank you for this valuable suggestion. We have addressed this by incorporating a section summarizing what is known in the field of epigenetics and how it should be integrated to address the key challenges in the manuscript.

Reviewer 1

Thanks for the revision & the modifications performed accordingly

Response

We would like to thank the reviewer for their comments

Reviewer 2 Report

Comments and Suggestions for Authors

Dear authors,

Thanks very much for your excellent detailed and comprehensive work

A few comments to be noted

1- The most important concern regarding the article is that it is extremely long. This length would have been important if every section added something to the context, but conversely. many paragraphs are just repetitions of the preceding ones using different words. This can be applied to all the sections of the manuscript. After finishing revision of the manuscript, I found that nearly all the marks made on the proof are comments on repeated paragraphs

2- Figure 1 can be omitted

3- Table 1 also should be omitted

4- Please add a table or a figure enumerating the four molecular types with the definition, genetic basics and the impact on treatment

Author Response

Comment 1

The most important concern regarding the article is that it is
extremely long. This length would have been important if every
section added something to the context, but conversely. many paragraphs
are just repetitions of the preceding ones using different words. This
can be applied to all the sections of the manuscript. After finishing
revision of the manuscript, I found that nearly all the marks made on
the proof are comments on repeated paragraphs.

Response

Larde sections repeating similar concepts in sections 4, 5 and 6 have been deleted

Comment  2

Figure 1 can be omitted

Response

A new revised version of figure 1 has been added

Comment 3

Table 1 also should be omitted

Response

The table has been removed

Comment 4

Please add a table or a figure enumerating the four molecular
types with the definition, genetic basics and the impact on treatment."

Response

A new revised version of figure3 with the definition, prognosis, genetic basis and treatment has been included in this figure

Reviewer 3 Report

Comments and Suggestions for Authors

Figure 1A: The legend for the graph is insufficient. Include “income” the descriptor for each line. Don’t force the reader to refer to the legend.

Figure 1 legend: “ajor”? use English text. “Showws” The heatmap grid below each map needs to be defined on the grid itself.

Cell of origin for each of the histotypes of Type II endometrial cacers should be discussed. For example, it is interesting the HGSOC shows essentially with 100% penetrance mutations in TP53. This links to the following statement in the manuscript “TP53 mutations are a hallmark of serous carcinoma, and these tumors often lack hormone receptor expression [15]”. How similar is the clear cell histotype to ovarian clear cell carcinoma i.e., at the molecular level?

I find a significant amount of redundancy in this manuscript. For example on page 7: 

The histological overlap between poorly differentiated EC subtypes, such as serous and high-grade endometrioid carcinomas, can blur the distinction between these categories. This overlap in morphology can lead to misclassification, which can impact treatment decisions and ultimately affect patient outcomes [7].

PDEECs often show overlapping histological features, making it difficult to differentiate

between subtypes based solely on morphology. This is particularly evident in highgrade tumors, where classification into distinct subtypes based on traditional histopathological criteria is frequently imprecise [20].

This section could be combined into a couple of sentences. This review is exceedingly long and would benefit considerably by a careful editing to reduce length and wordage.

Here’s another example, in my opinion:

“Sole reliance on histomorphological evaluation often results in incomplete diagnostic information, especially in tumors that present ambiguous or mixed features. Thus, additional techniques such as immunohistochemistry (IHC) have become crucial in achieving more precise classification of these tumors [2]. Targeted IHC panels are employed to distinguish between specific high-grade endometrial cancer subtypes [21] [22]. For example, the markers p53, p16, HNF-1β, ER/PR, and mismatch repair (MMR) proteins are commonly used to differentiate serous carcinoma from clear cell carcinoma [22]. Likewise, markers like ER, PR, napsin A, and AMACR aid in distinguishing endometrioid carcinoma from clear cell carcinoma… “ What about ARID1A?

The last paragraph of Section 2 is also dense with redundant statements and could be reduced to two sentences since it is essentially a recap of what was stated in the previous several paragraphs.

The authors might consider statements regarding DNA damage response pathways in the CNH vs CNL tumours.

This is the sort of statement that is repeated over and over again in the manuscript and doesn’t need to be; the reader gets the point:

“In cases where histomorphology is ambiguous, molecular profiling provides critical prognostic insights [32].”

“The genetic alterations provide insight into the mechanisms driving tumor development and offer potential avenues for targeted therapy. Deciphering the genomic landscape of PDEECs not only deepens the understanding of the biological underpinnings of these aggressive cancers but also opens the door for personalized therapeutic approaches [34].”

In this paragraph focussed on TP53, it is perhaps more interesting to provide molecular details regarding the function of mutated p53. For example, the vast majority of p53 mutations result in a long-lived protein relative to the wt protein which turns over rapidly and often provides a weak signal on IHC. It is well-established now, that mutated p53 has distinct functions from the wt protein and may form tetramers with the wt protein which can alter molecular properties on DNA.

“The loss of p53 function leads to genomic instability, a hallmark of PDEECs. Without

functional p53 to halt cell division in the presence of DNA damage, cells accumulate further mutations, which drive the progression and heterogeneity of the tumor [35]. Genomic instability contributes to both intra-tumor and inter-tumor heterogeneity, making treatment more challenging. Cells with TP53 mutations in PDEECs exhibit resistance to apoptosis,……….”

PTEN gene mutations: what type of mutations are these? Deactivating presumable. Are there hotspot mutations which kill the active site or are these truncating frame-shift mutations which result in loss of the protein. The same goes for PIK3CA mutations; how do these mutations results in a more active enzyme on which downstream targets?

The molecular pathways sections are to this reader, most relevant and important. The previous sections could be significantly shortened as this review is too long. There is a general lack of details throughout the manuscript. The manuscript is replete with generalities and little details i.e., experimental detail, reference to seminal literature describing the impact of important mutations on cancer cell behaviour and the best model systems with which to study this disease. 

There is little description, if any, of the surgical interventions typically used to treat these cancers

Importantly, it is surprising that the authors don’t mention the important side-effect of immune-suppression for many of the PI3K/AKT/mTOR pathway inhibitors which can have dire consequences such as life-threatening infections.

In summary, there is a great deal of relevant information in this review. The citation list alone will be useful to people in the field. I think that more detailed information on specific high impact points such as the potential combinations therapies with or without chemotherapy should have been included. Relevant model systems to study Type II endometrial cancers is also highly relevant. From perspective this review is overly verbose and requires a significant paring down to be concise and to the point with a significant removal of redundancies which is its biggest issue. There are far too may repeated sections and sentences especially between the initial sections of the review and sections 5 through to the end of the review. The authors obviously want this review to be comprehensive which should be applauded but to be digestible by the reader, relevance of each section should be challenged regarding other section in the manuscript. Do you really need the long pre-amble? Effective editorial work on this manuscript will be essential.

Most importantly, how is this manuscript different from:

Molefi T, Mabonga L, Hull R, Sebitloane M, Dlamini Z. From Genes to Clinical Practice: Exploring the Genomic Underpinnings of Endometrial Cancer. Cancers (Basel). 2025 Jan 20;17(2):320. doi: 10.3390/cancers17020320. PMID: 39858102; PMCID: PMC11763595.

Same authors and same subject matter; the above review is very similar to this manuscript.

Comments on the Quality of English Language

I have no serious concerns with English sentence structure or grammar. However, there is far too much in the way of redundant statements and information through the manuscript. It could easily be reduced to half its current length. Some of the gigantic diagrams could also be reduced in size.

Author Response

Reviewer 4
Regarding DNA damage response pathways in the CNH vs CNL tumors.
Response: We appreciate this insightful comment. To clarify, we have revised and discussed DNA damage response pathways in the CNH vs. CNL tumors in specific sections of the manuscript. This modification improves the clarity and accuracy of our findings.
This is the sort of statement that is repeated repeatedly in the manuscript and does not need to be; the reader gets the point:
“In cases where histomorphology is ambiguous, molecular profiling provides critical prognostic insights [32].”
“The genetic alterations provide insight into the mechanisms driving tumor development and offer potential avenues for targeted therapy. Deciphering the genomic landscape of PDEECs not only deepens the understanding of the biological underpinnings of these aggressive cancers but also opens the door for personalized therapeutic approaches [34].”
Response: We acknowledge this point and have modified the text accordingly.
In this paragraph focussed on TP53, it is perhaps more interesting to provide molecular details regarding the function of mutated p53. For example, the vast majority of p53 mutations result in a long-lived protein relative to the wt protein, which turns over rapidly and often provides a weak signal on IHC. It is well-established now, that mutated p53 has distinct functions from the wt protein and may form tetramers with the wt protein, which can alter molecular properties on DNA.
“The loss of p53 function leads to genomic instability, a hallmark of PDEECs. Without functional p53 to halt cell division in the presence of DNA damage, cells accumulate further
mutations, that drive the progression and heterogeneity of the tumor [35]. Genomic instability contributes to both intra-tumor and inter-tumor heterogeneity, making treatment more challenging. Cells with TP53 mutations in PDEECs exhibit resistance to apoptosis,……….”
Response: We appreciate this insightful comment. To clarify this, we have provided molecular details regarding the function of mutated p53. This modification improves the clarity and accuracy of our findings.
PTEN gene mutations: what type of mutations are these? presumable. Are there hotspot mutations which kill the active site or are these truncating frame-shift mutations which result in loss of the protein. The same goes for PIK3CA mutations; how do these mutations results in a more active enzyme on which downstream targets?
Response: We acknowledge this point and have modified the text accordingly. The revised section now states the types and modus operandi of PTEN and PIK3CA mutations.
The molecular sections are to this reader, most relevant and important. The previous sections can be significantly shortened because as this review is too long. There is a general lack of details throughout the manuscript. The manuscript is replete with generalities and few/little details i.e., experimental detail, reference to seminal literature describing the impact of important mutations on cancer cell behavior, and the best model systems with which to study this disease.
Response: Thank you for this valuable suggestion. We have addressed this by adding experimental details, reference to seminal literature describing the impact of important mutations on cancer cell behavior, and the best model systems with which to study this disease.
There is little description, if any, of the surgical interventions typically used to treat these cancers
Response: We acknowledge this point and have incorporated a section on the surgical interventions typically used to treat PDEECs.
Importantly, it is surprising that the authors don’t mention the important side-effect of immune suppression for many of the PI3K/AKT/mTOR pathway inhibitors, which can have dire consequences, such as life-threatening infections.
Response: Thank you for your valuable comment. We appreciate your attention to this important aspect of PI3K/AKT/mTOR inhibitors. In response to your concern, we have added a new subsection titled "Immunosuppressive Effects of PI3K/AKT/mTOR Inhibitors" to Section 5.1 on the PI3K/AKT/mTOR Pathway. This new subsection addresses the significant immunosuppressive side effects associated with PI3K/AKT/mTOR pathway inhibitors, including the increased risk of life-threatening infections. We discuss the mechanisms underlying this immunosuppression, its clinical implications, and the challenges it presents in
balancing antitumor effects with potential risks. We believe that this addition provides a more comprehensive overview of the challenges associated with targeting the PI3K/AKT/mTOR pathway in PDEECs and addresses the important points you raised. We thank you for bringing this to our attention, as it significantly enhanced the clinical relevance and completeness of our review.Thank you for this valuable suggestion. We have addressed this by discussing the important side-effect of immune-suppression for many of the PI3K/AKT/mTOR pathway inhibitors which can have dire consequences such as life-threatening infections.
In summary, there is a great deal of relevant information in this review. The citation list alone will be useful to people in the field. I think that more detailed information on specific high impact points such as the potential combinations therapies with or without chemotherapy should have been included. Relevant model systems to study Type II endometrial cancers is also highly relevant. From perspective this review is overly verbose and requires a significant paring down to be concise and to the point with a significant removal of redundancies which is its biggest issue. There are far too may repeated sections and sentences especially between the initial sections of the review and sections 5 through to the end of the review. The authors obviously want this review to be comprehensive which should be applauded but to be digestible by the reader, relevance of each section should be challenged regarding other section in the manuscript. Do you really need the long pre-amble? Effective editorial work on this manuscript will be essential.
Response: We have endeavoured to remove repetitions and long explanations to shorten the review. We have also edited the introduction However in order to keep the review to a manageable length we have decided to omit further description of combination therapies beyond that which is already in the paper
Most importantly, how is this manuscript different from:
Molefi T, Mabonga L, Hull R, Sebitloane M, Dlamini Z. From Genes to Clinical Practice: Exploring the Genomic Underpinnings of Endometrial Cancer. Cancers (Basel). 2025 Jan 20;17(2):320. doi: 10.3390/cancers17020320. PMID: 39858102; PMCID: PMC11763595.
Same authors and same subject matter; the above review is very similar to this manuscript.
Response: We appreciate this insightful comment. The above highlighted manuscript dwells much on the significance of the genomic underpinnings to clinical practice whereas this manuscript focuses more on identifying key molecular subtypes and associated genetic mutations which are prevalent in aggressive variants, and how they are exploited towards protracted therapeutic remissions
Comments on the Quality of English Language
I have no serious concerns with English sentence structure or grammar. However, there is far too much in the way of redundant statements and information through the manuscript. It can easily be reduced to half its current length. Some of the gigantic diagrams could also be reduced in size.
Response: We appreciate this insightful comment.

Round 2

Reviewer 2 Report

Comments and Suggestions for Authors

Thanks for the revision & the modifications performed accordingly

Author Response

Reviewer2:

Thanks

Reviewer 3 Report

Comments and Suggestions for Authors

Just by example below, there are still sections of the manuscript that are redundant and could be much more concisely stated.

"In the loss of inhibition by p85, wild-type PI3K is tightly regulated by p85, which prevents spontaneous activation, and helical domain mutations (E542K, E545K) disrupt p85 binding, causing unrestrained PI3K activity [62, 65]."

The above sentence should read:

The p85  subunit helical domain mutations (E542K, E545K) prevents binding to PIK3CA leading to unrestrained PI3K activity and elevated AKT signalling leading to enhanced cancer cell proliferation and viability.

Comments on the Quality of English Language

This is an editorial decision by the journal to make. I think this extremely lengthy review would be significantly improved by reducing areas of redundancy. If you want to see it cited extensively, then it has to be more concise otherwise, in my opinion, people won't bother reading it.

Author Response

Comment 1

Just by example below, there are still sections of the manuscript that are redundant and could be much more concisely stated.

Response

Sections have been deleted especially common statements between sections 4, 5 and 6.

Comment 2

"In the loss of inhibition by p85, wild-type PI3K is tightly regulated by p85, which prevents spontaneous activation, and helical domain mutations (E542K, E545K) disrupt p85 binding, causing unrestrained PI3K activity [62, 65]."

The above sentence should read:

The p85  subunit helical domain mutations (E542K, E545K) prevents binding to PIK3CA leading to unrestrained PI3K activity and elevated AKT signalling leading to enhanced cancer cell proliferation and viability.

Response

The sentence has been corrected

Comment 1

Just by example below, there are still sections of the manuscript that are redundant and could be much more concisely stated.

Response

Sections have been deleted especially common statements between sections 4, 5 and 6.

Comment 2

"In the loss of inhibition by p85, wild-type PI3K is tightly regulated by p85, which prevents spontaneous activation, and helical domain mutations (E542K, E545K) disrupt p85 binding, causing unrestrained PI3K activity [62, 65]."

The above sentence should read:

The p85  subunit helical domain mutations (E542K, E545K) prevents binding to PIK3CA leading to unrestrained PI3K activity and elevated AKT signalling leading to enhanced cancer cell proliferation and viability.

Response

The sentence has been corrected
